# Probe-free optical chromatin deformation and measurement of differential mechanical properties in the nucleus

**Benjamin Seelbinder**[1,2], **Susan Wagner**[1,3]*, **Manavi Jain**[1,2], **Elena Erben**[1,2], **Sergei Klykov**[1,2], **Iliya Dimitrov Stoev**[1,2], **Venkat Raghavan Krishnaswamy**[1], **Moritz Kreysing**[1,2,3]

[1]Max Planck Institute of Molecular Cell Biology and Genetics, Dresden, Germany; [2]Centre for Systems Biology, Dresden, Germany; [3]Institute of Biological and Chemical Systems-Biological Information Processing, Karlsruhe Institute of Technology, Eggenstein-Leopoldshafen, Germany

**Abstract** The nucleus is highly organized to facilitate coordinated gene transcription. Measuring the rheological properties of the nucleus and its sub-compartments will be crucial to understand the principles underlying nuclear organization. Here, we show that strongly localized temperature gradients (approaching 1°C/µm) can lead to substantial intra-nuclear chromatin displacements (>1 µm), while nuclear area and lamina shape remain unaffected. Using particle image velocimetry (PIV), intra-nuclear displacement fields can be calculated and converted into spatio-temporally resolved maps of various strain components. Using this approach, we show that chromatin displacements are highly reversible, indicating that elastic contributions are dominant in maintaining nuclear organization on the time scale of seconds. In genetically inverted nuclei, centrally compacted heterochromatin displays high resistance to deformation, giving a rigid, solid-like appearance. Correlating spatially resolved strain maps with fluorescent reporters in conventional interphase nuclei reveals that various nuclear compartments possess distinct mechanical identities. Surprisingly, both densely and loosely packed chromatin showed high resistance to deformation, compared to medium dense chromatin. Equally, nucleoli display particularly high resistance and strong local anchoring to heterochromatin. Our results establish how localized temperature gradients can be used to drive nuclear compartments out of mechanical equilibrium to obtain spatial maps of their material responses.

*For correspondence: susan.wagner@kit.edu

## Editor's evaluation

Seelbinder et al. describe a valuable new method for perturbing chromatin in living cells by strongly localized temperature gradients. Solid analysis reveals that chromatin shows both elastic and viscous properties at the timescales of seconds, with heterochromatin showing solid-like properties. While some details of the nuclear response to local heating remain to be elucidated, the ability of the method to reveal local mechanics in vivo makes the approach likely to be of broad interest to both the cell biophysics and cell biology communities.

## Introduction

It is widely believed that the spatial organization of the nucleus is supported by and makes functional use of distinct material properties to support homeostatic function (*Cremer et al., 2020*; *Falk et al., 2019*; *Mirny and Dekker, 2022*), and that may be temporally adapted to facilitate cell cycle dynamics and differentiation (*Mittasch et al., 2020*; *Strom et al., 2021*; *Sun et al., 2018*) or cell migration

through confined spaces (*Denais et al., 2016*; *Irianto et al., 2017*; *Pfeifer et al., 2018*; *Shah et al., 2021*). Starting at the nanometer scale, molecular interactions are thought to give rise to the spatial organization of nuclear constituents that become visible at the micrometer scale. For example, the nucleus features membraneless organelles, such as the nucleolus, Cajal bodies, nuclear speckles, PML bodies, and others, which are comprised of RNA and proteins and are considered to be formed by liquid-liquid phase separation (LLPS) (*Zidovska, 2020a*; *Feric et al., 2016*). Most of the nucleus is occupied by chromatin, however, which is hierarchically organized into different compartments: (i) a few nucleosomes (5-20) loosely assembled into clutches, which further assemble into chromatin nano-domains, (ii) nanodomains are further grouped into local continuous gene clusters called topologically associated domains (TADs), and (iii) TADs from different loci group together to form two main compartments: active A-compartments and inactive B-compartments (*Jerkovic´ and Cavalli, 2021*; *Mirny and Dekker, 2022*; *Misteli, 2020*). Using data from chromatin conformation capturing assays (Hi-C), computer simulations suggest that strong interactions between B constituents together with weak interactions of A constituents drive the separation of compartments (*Falk et al., 2019*; *MacPherson et al., 2018*; *MacPherson et al., 2020*). Furthermore, condensed chromatin is thought to solidify with increasing strength of molecular interactions, yielding elastic rather than viscous responses (*Hansen et al., 2021*) and increase in euchromatin leads to softening of the nucleus (*Stephens et al., 2018*). Recent studies demonstrated that chromatin compaction by HP1 proteins results in phase-separated liquid condensates (*Sanulli et al., 2019*; *Keenen et al., 2021*). Hence, differences in interactions within compartments should also be directly measurable as a reflection of different material properties. Experimental characterization of the material properties of nuclear compartments will be crucial to understand spatial nuclear organization and its role in nuclear information processing. Despite great progress, an integrated physical picture of the nucleus is still missing.

A recent point of discussion has been whether chromatin behaves like a liquid or a solid (*Strickfaden et al., 2020*; *Zidovska, 2020b*). For chromatin it has been suggested that its material properties are predominantly liquid in line with the view that heterochromatin domains form through LLPS, facilitated through the binding to scaffold proteins, similar to nuclear organelles (*Gibson et al., 2019*; *Larson et al., 2017*; *Strom et al., 2017*). However, FRAP experiments in interphase nuclei that harbored fluorescently labeled chromatin observed that bleached hetero- or euchromatin regions did not recover their intensity after bleaching at the minute to hour scale (*Strickfaden et al., 2020*). Contrary to the earlier view, this indicates that chromatin cannot move freely and therefore behaves more like a solid. To reconcile the observed discrepancies, it has been suggested that chromatin, like other polymers, shows a more complex behavior that can be viscous, elastic, or viscoelastic depending on the time and length scales that are being probed and the energy-driven enzymatic activity in the environment (*Zidovska, 2020b*; *Zidovska et al., 2013*). For example, the liquid-like behavior of chromatin observed at the nanoscale (*Nozaki et al., 2017*) likely emerges from enzymatically driven processes such as transcription and loop extrusion (*Fudenberg et al., 2016*; *Golfier et al., 2020*). A high molecular mobility on the nanoscale might still be locally constrained and prevent large-scale rearrangements of chromatin. Hence, observing liquid-like behavior at the nanoscale does not have to be in contradiction with solid-like behavior at the micron scale.

One functional adaptation of nuclear architecture is the central compaction of heterochromatin in rod cells, termed nuclear inversion (*Solovei et al., 2009*). Nuclear inversion is triggered by LBR down-regulation (*Solovei et al., 2013*) and describes the successive fusion of chromocenters during terminal stages of retinal development, which leads to improved contrast sensitivity under low light conditions (*Subramanian et al., 2021*). Motivated by these findings, it has been suggested that heterochromatin cohesion drives nuclear inversion, as well as the separation of hetero- and euchromatin in conventional interphase nuclei (*Falk et al., 2019*). Yet, it remains an open question which material properties compacted heterochromatin adapts, how more heterochromatin is mechanically integrated into the nucleoplasm, and how chromatin interfaces with other nuclear compartments. To better understand chromatin organization across scales, new experimental approaches are needed to measure material properties in living cells, ideally spatially resolved and dynamically.

A wide range of different complementary techniques have been proposed to infer compartment interactions or nuclear material properties. At the nanoscale, Hi-C-based methods have been used to map the spatial interactions between different chromatin compartments (*Belaghzal et al., 2021*; *Falk et al., 2019*; *Lieberman-Aiden et al., 2009*) and reconstitute the genomic 3D organization in

silico (*Stevens et al., 2017*). While Hi-C reports on genome interactions and allows to infer the spatial organization to a very good degree, Hi-C methods themselves do not capture dynamic processes or facilitate making predictions about the material properties, without being complemented by other techniques.

To acquire dynamic data of chromatin motion at the mesoscale (0.01–1 μm), passive micro-rheology approaches can be used to infer material properties from the spatio-temporal dynamics of discernable features using video microscopy in conjunction with tracking algorithms (*Armiger et al., 2018*; *Eshghi et al., 2021*; *Herráez-Aguilar et al., 2020*; *Nozaki et al., 2017*; *Zidovska et al., 2013*). These methods are successful in gaining insights into apparent material properties non-invasively; however, their use is limited to short time scales during which thermal fluctuation dominates over active processes. At larger time scales, motion appears to be largely driven by active ATP-dependent processes and material properties cannot be quantified anymore by assuming thermal motion as the driver (*Guo et al., 2014*; *Zidovska et al., 2013*). For example, chromatin shows ATP-dependent coherent motion on time scales above 1 s (*Zidovska et al., 2013*).

To overcome the limitations of passive micro-rheology in energy-consuming materials, active micro-rheology approaches have been developed that use an external stimulus to drive materials out of their mechanical equilibrium. Methods to test nuclear material properties include AFM (*Sheng et al., 2014*), micropipette aspiration (*Dahl et al., 2004*; *Dahl et al., 2005*; *Davidson et al., 2019*; *Pajerowski et al., 2007*), micromanipulation (*Stephens et al., 2017*; *Stephens et al., 2018*), non-invasive techniques using natural and artificial probes (*Caragine et al., 2018*; *Caragine et al., 2021*; *Lee et al., 2021*; *Shin et al., 2018*), magnetic beads (*Guilluy et al., 2014*; *Keizer et al., 2022*), and membrane stretch devices (*Schürmann et al., 2016*; *Seelbinder et al., 2020*). A difficulty, however, is that the nuclear lamina, an outer stiff shell that surrounds and protects the nucleus, is believed to be at least 10× stiffer than chromatin and would therefore mask the internal material properties when probed from the outside (*Harada et al., 2014*; *Isermann and Lammerding, 2013*), while chromatin governs response to small extensions (<3 μm) and the lamina to larger extensions (*Stephens et al., 2017*). These methods provide a good understanding of the mechanics of the nucleus as an integrated whole, but lack spatial resolution. Recently, the injection of magnetic beads into live nuclei allowed for the estimation of local material properties inside the nucleus, but spatial control of the bead location is still limited (*Keizer et al., 2022*).

Spatially resolved maps of material properties can be obtained by Brillouin microscopy (BM) (*Brillouin, 1922*; *Prevedel et al., 2019*; *Scarcelli and Yun, 2008*), by quantifying Raman shifts of scattered photons. BM has been used to map mechanical properties in zebrafish (*Sánchez-Iranzo et al., 2020*) and living cell nuclei (*Zhang et al., 2020*; *Zhang et al., 2017*). While it is a powerful method, BM measures the mechanics at very short time scales (sub-nanosecond) due to its reliance on acoustic waves in the GHz range. Highly attractive would therefore be a micro-rheology method that permits probe-free, spatially resolved mapping of material properties on physiologically relevant time scales, while leveraging the conceptual advantages of active perturbations.

Here, we report on a novel approach that utilizes highly localized temperature gradients to displace and strain chromatin inside the nucleus. Since displacements can reach hundreds of nanometers to few microns, local displacements can be quantified through tracking algorithms. As temperature stimuli pervade the stiff nuclear lamina, chromatin motion occurs without disturbing nuclear size or shape. Using this method, we observed that interphase chromatin displays highly reversible, visco-elastic behavior in response to deformation with a characteristic time of $\tau$ ~1 s. We find that material properties are spatially distinct for different compartments. The nucleolus, in particular, shows high mechanical resilience to deformation, but significant adhesion to surrounding chromatin.

Together, our results showcase the utility of this new approach to actively probe nuclear material properties in a spatially and temporally resolved manner, enabling to assign material responses to biochemical identity of compartments and gain new insights into the mechanics of compartment interfaces.

## Results

### Engineered temperature gradients facilitate controlled chromatin deformation in living cells

To generate a highly localized temperature gradient, a heating IR laser beam ( $\lambda$ =1455 nm) was focused and rapidly scanned (1 kHz) along a line (*Figure 1a*). Temperature profiles for various laser intensities (low, medium, high) were measured using the temperature-sensitive dye rhodamine B at a chamber temperature of 36°C (*Figure 1b–c*, *Figure 1—figure supplement 1a*). The average temperature increase inside the nucleus could further be confirmed via temperature-sensitive mCherry-H2b, which corresponded well to rhodamine measurements averaged along a typical nuclear length of 20 μm (*Figure 1d–e*, *Figure 1—figure supplement 1b*). Unless otherwise indicated, experiments were run at medium laser intensity resulting in an average temperature increase of 2°C inside the nucleus.

When placing the heating stimulus in close proximity to the nucleus, we observed chromatin motion down the temperature gradient as visualized by GFP-H2b in NIH-3T3 cells (*Figure 1f*, *Figure 1—videos 1; 2*). To exclude the possibility that an absolute temperature increase above 36°C is the driver of the observed chromatin movement, rather than the induced temperature gradient, we repeated our experiments at an ambient temperature of 30°C. As with an ambient temperature of 36°C, we observed chromatin motion upon the heating stimulus down the temperature gradient (*Figure 1—figure supplement 1c–e*).

The laser stimulus did not trigger a substantial stress response (*Figure 1—figure supplement 1f–h*). Formation of dynamic stress granules (SGs) is a typical and reversable response of cells to mitigate several kinds of stress (*Hofmann et al., 2021*). The protein G3BP1, for example, is a marker of SGs. Under normal conditions, G3BP1 is distributed in the cytoplasm, but accumulates into granules when a cell experiences a stress (*Hofmann et al., 2021*), such as short treatment with thapsigargin (*Sidrauski et al., 2015*; *Figure 1—figure supplement 1g*). Compared to the SG formation upon treatment with thapsigargin, we observed only minor formation of SGs if at all (*Figure 1—figure supplement 1h*). Within 15 min after a typical laser stimulus of 10 s, in seven repetitions we observed either no SG formation (three times), the formation of a few SGs (two times) or stronger SG formation (two times), but still to a lower extend compared to the positive control of thapsigargin treatment (*Figure 1—figure supplement 1h*). We concluded that the laser stimulus alone is not sufficient to trigger a stress response, although, additional stresses caused by the experimental setup, for example, mounting of the cells inside the thin chambers, can in sum lead to SG formation in some cases. Yet, in near half of the cases, our perturbations did not trigger any SG formation. This indicates that the laser-induced chromatin motion does not necessarily evoke a stress response. However, some stress response was observed in some cases. Yet, stress responses are typically mounted within minutes. The mechanical response to the laser stimulus, the chromatin movement, can be seen independent as it is much faster. Therefore, mechanical properties of the chromatin movement are independent of a later stress response. To dissect the occasional stress response, further studies would be needed, which could benefit from decoupling of laser heating and the induction of temperature gradients (*Minopoli et al., 2023*).

To better visualize time-dependent chromatin displacements, we generated image difference stacks by subtracting the first image from the image stack (*Figure 1f*). Analyzing the average image difference over time, we demonstrated that chromatin movement is instantaneous and largely reversible (*Figure 1g*). Further, image difference dynamics followed an exponential trend that could be fitted to a simple viscoelastic material model (Kelvin-Voigt), with characteristic times $\tau$ on the scale of seconds. In the absence of temperature stimulation, we only observed small changes in the image difference stack that likely reflect the spontaneous coherent motion of chromatin as reported before (*Zidovska et al., 2013*) and might account for some of the residual differences after perturbations.

### Strongly localized temperature gradients drive intra-nuclear chromatin displacement without affecting nuclear shape

To better characterize the observed chromatin displacement away from an applied temperature gradient, we tracked large intra-nuclear features during the perturbation. Large organelles that exclude chromatin, such as the nucleolus, show up as dark pockets (features) in GFP-H2b images and are well suited to track intra-nuclear movement (*Figure 2a*). Kymograph analysis of chromatin-void features

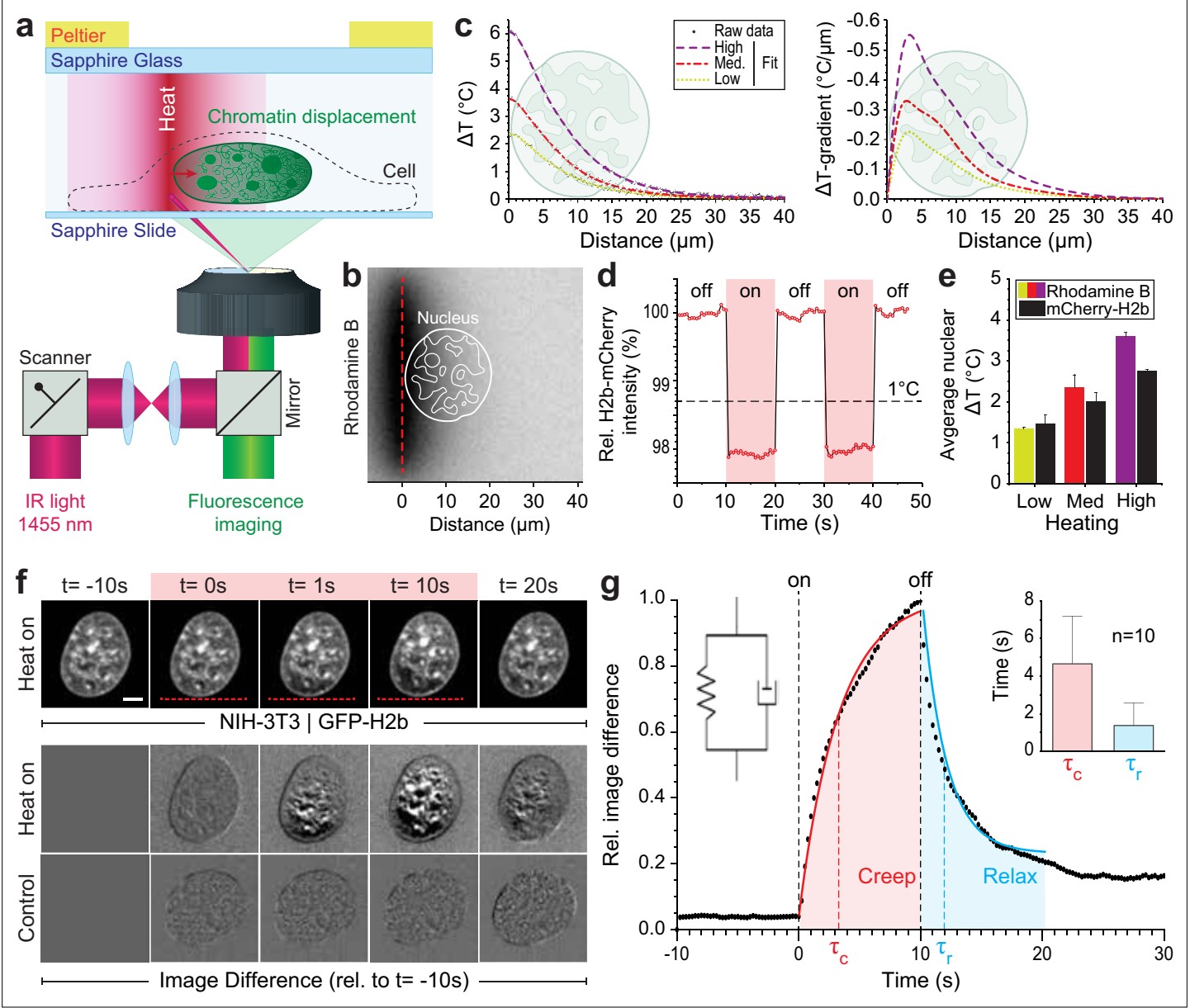

**Figure 1.** Engineered temperature gradients facilitate controlled chromatin deformation in living cells. (**a**) Overview of the microscopy setup. An IR laser is scanned along a line to generate a temperature gradient perpendicular to the scanning line. Cells are cultured in temperature chambers that maintain a reservoir temperature of 36°C via Peltier elements. Sapphire, rather than glass, was used due to its high heat transfer coefficient. Shown is the side view of chamber. (**b**) The temperature-sensitive dye rhodamine B can be used to visualize temperature gradients during laser scanning. The red dotted line indicates the laser scan path and outlines of an average size nucleus are superimposed in white. (**c**) Temperature profile and gradient were quantified perpendicular to the laser scan path via rhodamine B for three different laser intensities (low, medium, and high). Raw data of temperature measurements were fitted and differentiated to achieve noise-robust estimates of the temperature gradients. See *Figure 1—figure supplement 1* for dye calibration. (**d**) Confirmation of instantaneous temperature changes inside the nucleus, at low laser intensity, via relative quantum efficiency measurements of mCherry-H2b, which reduces ~1.3% for each 1°C heating. See *Figure 1—figure supplement 1* for calibration. (**e**) Comparison of thermometry results measured inside the nucleus via mCherry-H2b and inside the chamber via rhodamine B. Chamber temperature was averaged along 20 µm, reflecting the average size of a nucleus, to compare both measurements directly; n=5, error = STD. (**f**) Top: Response of NIH-3T3 nuclei to an applied temperature gradient at medium laser intensity. Bottom: Image difference analysis of the same data and controls, indicating mesoscopic, partially reversible chromatin displacements due to temperature stimulation; scale = 5 µm. See also *Figure 1—videos 1; 2*. (**g**) Temporal analysis of image differences shows that chromatin rearrangements follow a simple viscoelastic material model by Kelvin-Voigt (inset), and have characteristic times ($\tau_c$ - creep or retardation time, $\tau_r$ - relaxation time) on the time scale of seconds; n=10, error = STD. The Kelvin-Voigt model consists of a purely elastic spring and a purely viscous dashpot arranged in parallel, with uniform distribution of strain. There is no possibility for the spring and dashpot to expand independently leading to a typical creep-recovery response.

*Figure 1 continued on next page*

*Figure 1 continued*

The online version of this article includes the following video and figure supplement(s) for figure 1:

**Figure supplement 1.** Calibration of dyes for temperature measurements, temperature impact, cell viability, and stress response.

**Figure 1—video 1.** Video microscopy of temperature stimulated and control nuclei, corresponding to image difference analysis in *Figure 1*.
https://elifesciences.org/articles/76421/figures#fig1video1

**Figure 1—video 2.** Video microscopy of temperature stimulated and control nuclei, corresponding to image difference analysis in *Figure 1*.
https://elifesciences.org/articles/76421/figures#fig1video2

---

confirm instantaneous directed motion with amplitudes of hundreds of nanometers to a few microns, with reversible asymptotic exponential dynamics (*Figure 2b–c*). Furthermore, centroid tracking of prominent features reveal that displacements are larger closer to the temperature stimulus.

Displacements appear to be restricted to the inside of the nucleus, as the centroid of the nucleus and nuclear area only marginally changed (~1%) during temperature stimulation (*Figure 2d–e*, *Figure 2— video 1*). Specifically, quantifying the change in nuclear geometry by measuring the distance of the nuclear border to the nuclear center before (t=0 s), during (t=10 s), and after (t=20 s) temperature stimulation, we further validated that the deviations in nuclear shape were on the order of 100 nm (*Figure 2f–g*), similar to displacements of the nuclear centroid. The constant nuclear dimensions likely reflect the dominating stiffness of the nuclear laminar. Additionally, the constant volume of the nucleus distinguishes our findings from the nuclear swelling that was observed during bulk heating of isolated nuclei (*Chan et al., 2017*).

To summarize our methodological advancement to this point, we found that strongly localized temperature gradients with an absolute temperature increase below 2°C cause micron-scale displacements of chromatin and chromatin void organelles inside the nucleus without changing the nuclear area or disturbing the nuclear border.

## Centrally compacted heterochromatin behaves as an elastically suspended solid in a genetically induced nuclear inversion model

Tethering of chromatin to the nuclear border (perinuclear chromatin) is an important mechanism that shapes global chromatin structure (*Guelen et al., 2008*). For example, detachment of chromatin from the nuclear envelope by downregulation of lamin A/C and lamin B receptor leads to the inversion of the conventional chromatin architecture in murine photoreceptor cells with nuclei displaying a condensed heterochromatin cluster in the center (*Solovei et al., 2013*; *Solovei et al., 2009*). This change in nuclear organization has functional implications even beyond gene expression control, as it serves to improve nocturnal vision in mice (*Subramanian et al., 2021*; *Subramanian et al., 2019*). Due to its characteristic organization, inverted nuclei have become a prominent model to study heterochromatin formation and global genome organization (*Solovei et al., 2009*).

By overexpressing Casz1 (a zinc finger transcription factor) in NIH-3T3 cells, as shown before (*MacPherson et al., 2020*), we were able to induce chromatin inversion that, in some instances, leads to the formation of a single large central heterochromatin cluster (CHC), reminiscent of the organization of photoreceptor nuclei in mice (*Figure 3a*). Inverted nuclei present an interesting model for studying chromatin organization, specifically with respect to the material properties associated with heterochromatin formation. We observed a highly reversible displacement of the CHC upon temperature stimulation in videos and kymographs (*Figure 3b*, *Figure 3—video 1*), which was further verified by tracking of the CHC centroid (*Figure 3c*). At the same time, area and shape of the CHC remained constant during stimulation, indicating strong resistance to deformation and hence dominantly solid-like behavior (*Figure 3d*).

The dynamic displacement again fitted well to a simple viscoelastic Kelvin-Voigt model with characteristic times of $\tau$ =0.74 s during the creep and $\tau$ =0.92 s during the relaxation phase (*Figure 3c*). Since the CHC moved in its entirety and showed little deformation, this dynamic likely reflects the properties of chromatin fibers that span radially from the CHC to the nuclear border (*Figure 3e*). By measuring the initial length of the fibers at rest ($L_0$~3 μm) and assuming that the CHC centroid displacement was similar to the extension of fibers (ΔL), the strain of chromatin fibers was estimated to be up to 15% (*Figure 3c*, right axis). In contrast, the bulk strain of the CHC was below 1%, indicating that the nucleus features heterogeneous compartment-specific material properties.

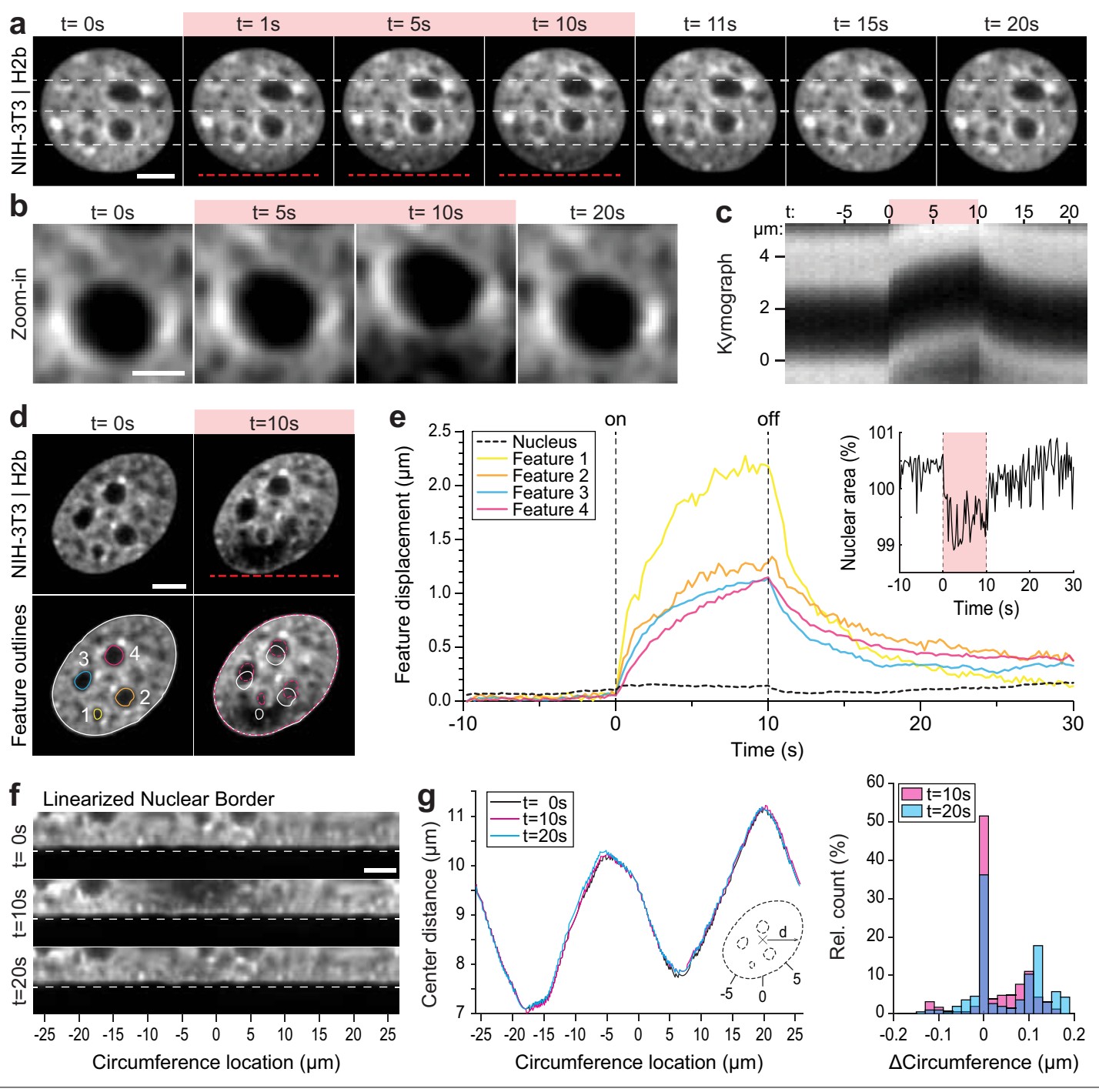

**Figure 2.** Quantification of intra-nuclear chromatins displacement and absence of changes in nuclear shape during temperature stimulation. (**a**) Chromatin displacement over time during 10 s temperature stimulation experiments in NIH-3T3 nuclei expressing H2b-GFP; scale = 5 µm. (**b–c**) Detailed view of one chromatin void feature. Kymograph analysis quantifying time and length scale shows largely reversible motion on the order of microns; scale = 5 µm. (**d–e**) Segmentation and tracking of intra-nuclear features show their gradual, heterogenous displacements of up to 2 µm, while nuclear area remains largely constant. See also *Figure 2—video 1*; scale = 5 µm. (**f**) Close-up view of the linearized nuclear border of the nucleus shown in (**d**) further indicates that the border remains static during and after temperature stimulation; scale = 5 µm. (**g**) Quantification of the distance of the nuclear border, with respect to the nuclear center at t=0 s, of the nucleus shown in (**d**) reveals that the distortions of the nuclear shape are less than 200 nm during temperature stimulation.

The online version of this article includes the following video for figure 2:

**Figure 2—video 1.** Video microscopy of temperature stimulated nuclei, corresponding to feature tracking analysis in *Figure 2*.
https://elifesciences.org/articles/76421/figures#fig2video1

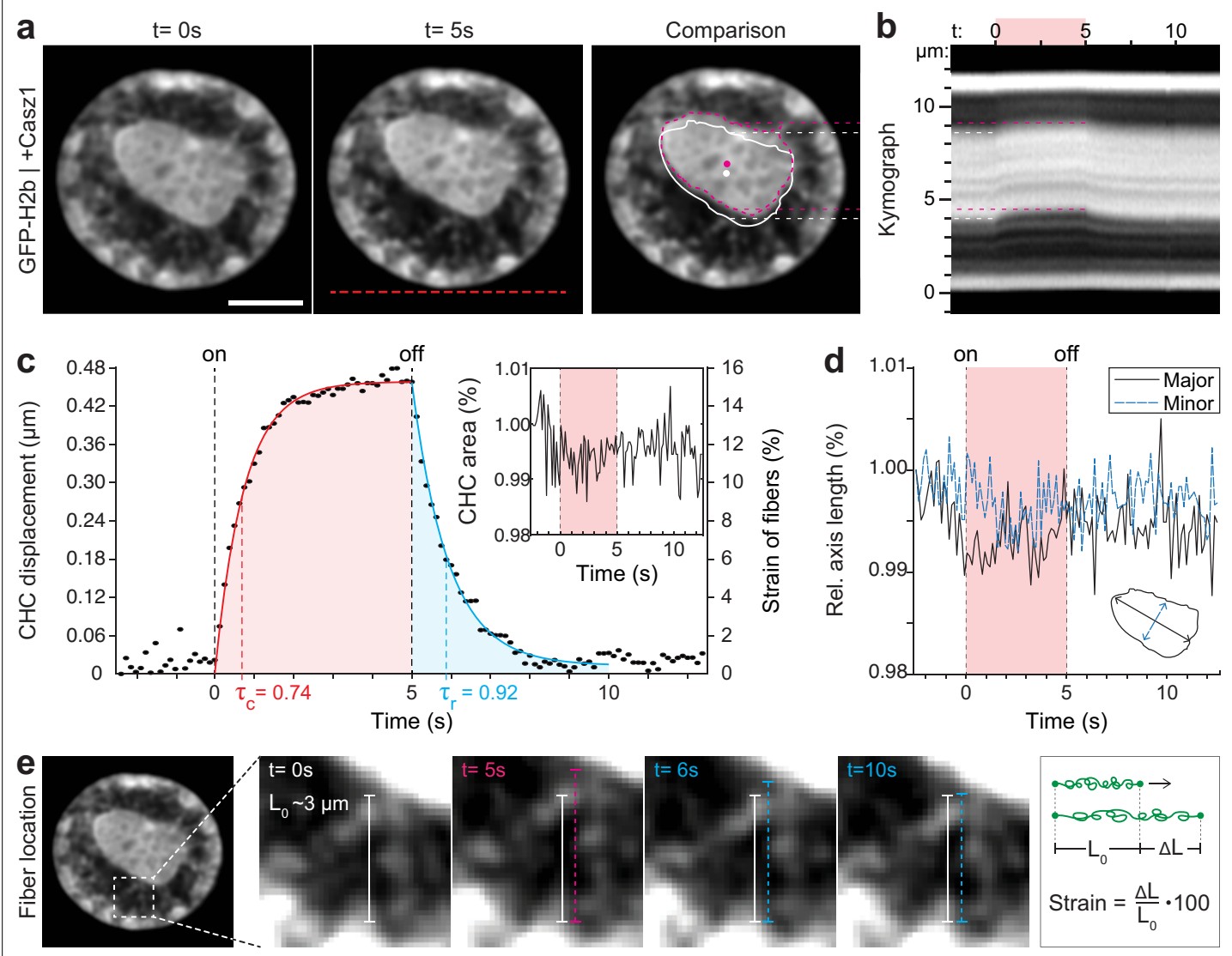

**Figure 3.** Quantification of movement and shape changes of centrally compacted chromatin during temperature stimulation in a genetically induced nuclear inversion model. (**a**) NIH-3T3 cells expressing H2b-GFP displayed an inverted chromatin organization after transfection with Casz1. The resulting central heterochromatin cluster (CHC) shows a significant displacement during temperature stimulation. See also *Figure 3—video 1*; scale = 5 µm. (**b**) Kymograph analysis quantifying time and length scale shows reversible submicron scale motion of the CHC. (**c–d**) Detailed analysis of the CHC centroid verifies that its motion is characterized by a fast and reversible displacement during temperature stimulation. However, the CHC appears resistant to deformation as it shows little change in area and major and minor axis length. (**e**) Close-up view of H2b-positive chromatin fibers straining during temperature stimulation. The cartoon on the right depicts the concept of strain. Based on their initial length $L_0$ ~3 µm, the estimated fiber strain is indicated in (**c**) on the right axis.

The online version of this article includes the following video for figure 3:

**Figure 3—video 1.** Video microscopy of temperature stimulated inverted nuclei, corresponding to analysis in *Figure 3*.

https://elifesciences.org/articles/76421/figures#fig3video1

## Spatially resolved strain maps reveal distinct mechanical properties of nuclear sub-compartments

Analyzing the chromatin displacement in inverted nuclei provided further evidence that the nucleus features spatially distinct heterogenous nuclear material properties. Therefore, we asked if we could map intra-nuclear strain more generally in conventional interphase nuclei in order to correlate local material identities with biological function. To this end, we used particle image velocimetry (PIV) (*Sveen, 2004*; *Zidovska et al., 2013*) to generate spatial displacement maps during temperature stimulation

using the frame at t=0 s as reference (*Figure 4a*, *Figure 4—figure supplement 1*). To further estimate local intra-nuclear deformation, strain maps were calculated from displacement maps. Shown here are local volumetric changes (hydrostatic strain) and orthogonal displacements (shear strain). Absolute nuclear strain magnitudes, integrated over the whole nucleus, displayed asymptotic exponential dynamics similar to tracked features before (*Figure 4b*). Averaging non-absolute (total) hydrostatic strains over the whole nucleus, where positive values (extension) and negative values (compression) can cancel each other, results in a line close to 0%, which is congruent with our observation that there is little change in overall nuclear area during temperature stimulations.

Using mCherry-H2b intensities as a proxy for chromatin density, we generated discretized maps of seven nuclear compartments (*Cremer et al., 2020*) of equal volume (*Figure 4c*). An additional dye was used to identify the nucleolus. Segmented fluorescence maps were then spatially correlated with strain maps to investigate whether there are differences in compartment behavior during temperature-induced chromatin displacements. Surprisingly, analysis of absolute hydrostatic strains revealed a non-linear relationship over chromatin densities, with most dense (C7) and most light packed regions (C1) experiencing the least, and medium dense regions (C4) the highest change in volume (*Figure 4d–e*, *Figure 4—figure supplement 2*). Despite similar propensities in volume change, analysis of total (non-absolute) hydrostatic strains further showed that lightly packed euchromatin bins (C1–3) tend to be compressed, while heterochromatin bins (C4–7) tend to be extended (*Figure 4f*). This likely reflected the inability of densely compacted chromatin to be further compacted, while loosely packed euchromatin seems to act as a mechanical buffer for decompaction inside a conserved volume.

The nucleolus is considered to be a liquid condensate that, despite lacking a membrane, maintains its integrity through LLPS (*Lafontaine et al., 2021*; *Strom and Brangwynne, 2019*). Surprisingly, nucleoli showed high mechanical resilience during temperature-induced chromatin displacement, as we measured only half the amount of absolute hydrostatic strain compared to overall chromatin (7.2% vs 13.5%) and a third less compared to dense chromatin regions (C7, 7.2% vs 10.4%) that are frequently found adjacent to nucleoli.

## Immobile nucleoli provide a model case to study chromatin-nucleoli and chromatin-chromatin interactions

The nucleolus is formed by nucleolar organizing regions that contain tandem copies of ribosomal DNA (*Bersaglieri and Santoro, 2019*). These regions form a characteristic ring of condensed chromatin around the nucleolus, referred to as perinucleolar chromatin. In contrast to the lamina, the molecular mechanisms of chromatin-nucleoli tethering are not well understood (*Mirny and Dekker, 2022*). We observed that the nucleus showed high mechanical resilience to deformation. Moreover, in some cases, we observed that the nucleoli remained largely static during our stimulations with chromatin appearing to flow around it (*Figure 5a*, *Figure 5—video 1*). To gain more insights into the way the nucleolus is mechanically embedded in nucleoplasm, we quantified the displacement of nucleoli and of adjacent 4 pixel thick (~0.5 μm) perinuclear regions with a distance of 0–0.5 μm (PC1), 0.5–1.0 μm (PC2), and 1.0–1.5 μm (PC3) from the nucleoli border. Quantification verified that nucleoli (NLL), which appeared static during temperature-induced chromatin displacement, moved significantly less compared to the nuclear average (NUC, *Figure 5b*). While chromatin appears to flow around nucleoli in videos and PIV displacement maps, detailed analysis showed that the displacement of the closest regions (PC1) was not significantly higher than that of nucleoli (*Figure 5c*). Displacements increased successively between the perinucleolar regions, but were still distinctly lower in PC3 compared to average chromatin displacements (NUC). This further suggested that, first, perinucleolar chromatin has a strong association with the liquid interface of the nucleoli border and, second, that the chromatin network is highly interwoven with signatures of continuous interaction on the scale of microns. As a control, velocity gradient does not occur around mobile nucleoli (*Figure 5—figure supplement 1*).

## Discussion

We demonstrated that strongly localized temperature gradients can move chromatin inside cell nuclei, thereby providing a complementary approach to existing methods to gain detailed new insight into the spatial organization of the nucleus.

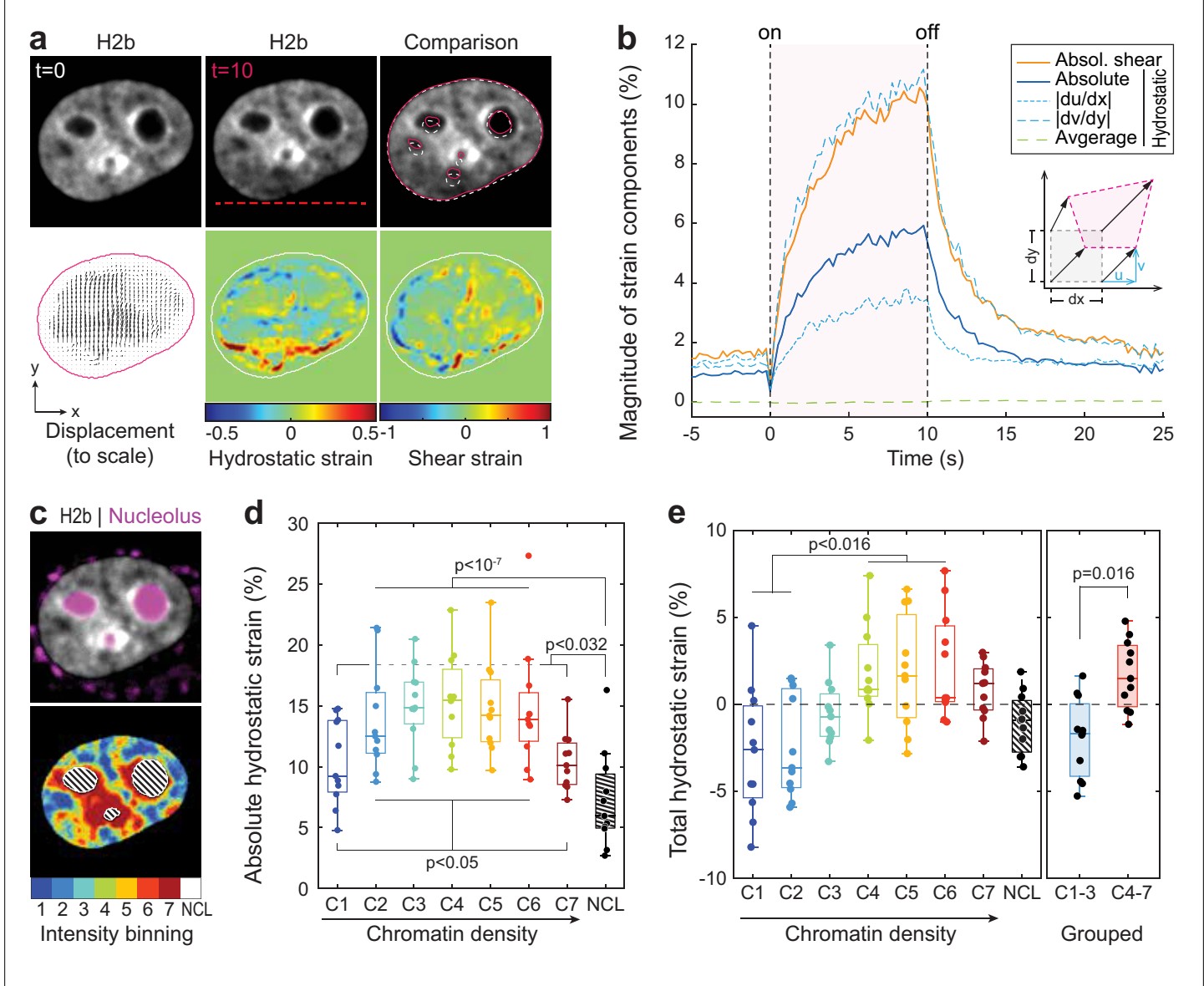

**Figure 4.** Differential strain measurement within chromatin compartments during temperature stimulation using spatially resolved strain maps. (**a**) NIH-3T3 cells expressing mCherry-H2b were recorded during temperature stimulation. Spatially resolved displacement maps were generated from image stacks via particle image velocimetry (PIV), using t=0 s as undeformed reference frame. From this, hydrostatic and shear strain maps were further calculated. Strains are shown here as relative values (1=100%). Displacement map is shown at half density, see *Figure 4—figure supplement 1* for full density and large-scale strain maps; scale = 5 µm. (**b**) Magnitude and temporal evolution of strain components. The inset cartoon depicts the concept of different strain types as a measure of local deformation. (**c**) Nuclei were segmented into seven equi-volumetric chromatin compartments of different density inferred by mCherry-H2b intensity. Nucleoli were further detected using Cytopainter live stains. A region of 6 pixels (~0.74 µm) away from the nuclear border was cut off to exclude low displacements close to the nuclear lamina. (**d**) Magnitudes of absolute hydrostatic strains (the sign of the strain is not considered, meaning positive values [extension] and negative values [compression] are added up) were locally evaluated for different chromatin densities by combining strain maps with compartment maps. Shown is the averaged absolute hydrostatic strain for each compartment measured at peak deformation (t=10 s) for n=11 nuclei. Boxplots depict the 25–75 percentile with whiskers spanning the full data range excluding outliers (>3× STD). Statistics via one-way ANOVA with Tukey HSD. (**e**) Local analysis of averaged total hydrostatic strains (the sign of the strain is considered meaning positive values [extension] and negative values [compression] cancel each other) of n=11 nuclei showing that lightly packed chromatin bins (C1–3) are preferentially compressed while densely packed chromatin bins are extended. Statistics via one-way ANOVA with Tukey HSD for all groups and two-tailed t-test for binned groups.

The online version of this article includes the following figure supplement(s) for figure 4:

**Figure supplement 1.** Displacement and strain maps and slight stiffening of the chromatin after temperature stimulation-induced chromatin displacement.

*Figure 4 continued on next page*

*Figure 4 continued*

**Figure supplement 2.** Hydrostatic strain over time for different nuclear compartments of a single nucleus.

We have shown that chromatin motion down the laser-induced temperature gradient occurs also at ambient temperatures much lower than 36°C, suggesting that the driver of the chromatin motion is the temperature gradient rather than the absolute temperature. From a rigorous physics point of view, one should note that a temperature gradient has a direction (is vectorial) which was consistently observed to be parallel to the observed motion of chromatin. The mere rise of temperature constitutes a rise of a scalar quantity, which does not provide a direction that could explain the directed motion of chromatin. Hence, it should be noted that only the gradient and not the absolute rise in temperature falls into a class of symmetries that is suitable to account for the observed effect of chromatin motion.

Concerning the slight and short-lived temperature increase above 36°C during our perturbation, we would like to mention that the range of temperature fluctuations that naturally occur within tissues and cells of organisms or that cells experience during common experimental practices are much wider than commonly assumed. For example, the arguable most temperature-sensitive cells are fertilized human egg cells, and considerable efforts are made to ensure highest temperature stability. Yet, the actual temperature in hoods and on microscope surfaces is in average more than 1.3°C different from the displayed temperature (*Palmer et al., 2019*). Also, core temperature in mice is not really constant but may vary by up to 5°C depending on ambient temperature (*Kaplan and Leveille, 1974*). Bovine embryonic development in vitro was unaffected by temperatures of up to 40°C, while deleterious effects were observable at a temperature of 41°C (*Rivera and Hansen, 2001*). This all is against the common view that only 37°C are seen as physiological temperature for mammalian cells.

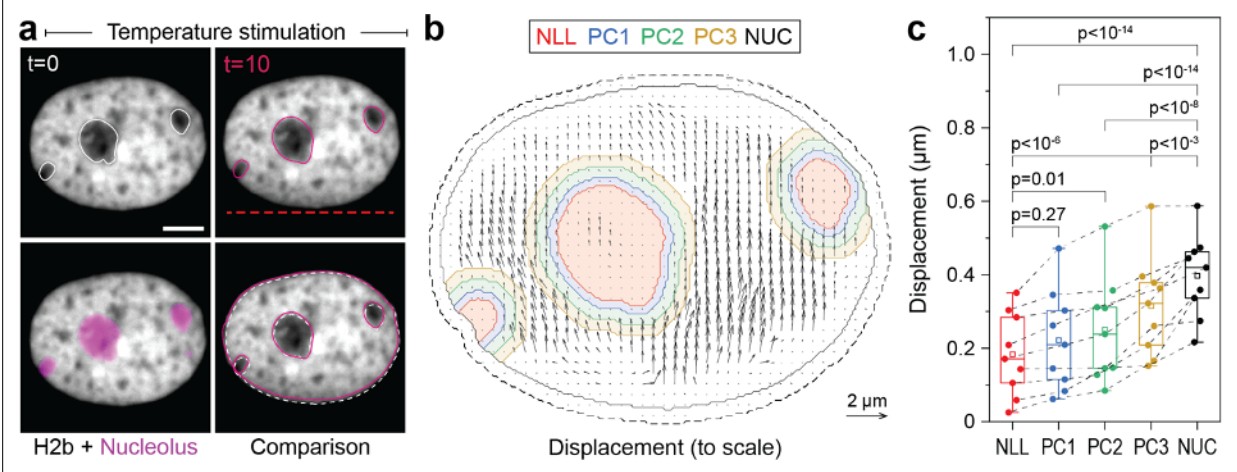

**Figure 5.** Measurement of chromatin displacements around static nucleoli reveal robust binding of perinuclear chromatin to the nucleolus as well as chromatin network effects. (**a**) NIH-3T3 cells expressing H2b-GFP were stained with a nucleoli live stain and recorded during temperature-induced chromatin displacement. Outlines represent detected nucleoli. Shown is an example in which nucleoli displayed little displacement, with chromatin flowing around the nucleoli like an obstacle. Red dotted line indicates position of laser scan path. See also *Figure 5—video 1*; scale = 5 µm. (**b**) Displacement maps, derived via particle image velocimetry (PIV), show chromatin motion around an immobile nucleolus. Indicated are the outlines of the nucleolus (NLL) and perinucleolar chromatin shells with a distance of 0–0.5 µm (PC1), 0.5–1 µm (PC2), and 1–1.5 µm (PC3) as well as the nuclear border (NUC). Low displacements close to the nuclear lamina (6 pixels ~0.74 µm, dotted line) were excluded to better reflect internal chromatin motion. Displacement maps are shown at half density. (**c**) Displacements for the nucleolus (NLL), perinuclear chromatin shells (PC1–3), and the nucleus as a whole (NUC) were quantified for n=9 different nuclei that displayed static nucleoli. See also *Figure 5—figure supplement 1* for the cases of moving nucleoli as a comparison. Statistics via one-way ANOVA with Tukey HSD.

The online version of this article includes the following video and figure supplement(s) for figure 5:

**Figure supplement 1.** Displacement analysis of moving nucleoli as comparison for static nucleoli. (**a**) NIH-3T3 cells expressing H2b-GFP were stained with a nucleoli live stain and recorded during temperature-induced chromatin displacement.

**Figure 5—video 1.** Video microscopy of temperature stimulated nuclei, showing a case of static nucleoli and corresponding to analysis in *Figure 5*. https://elifesciences.org/articles/76421/figures#fig5video1

Although we cannot exclude that rates for biochemical reactions within the cell might be altered moderately due to the change in temperature, the change of local concentrations of products and educts will be marginal due to time scales of only few seconds and the narrow range of temperature gradient of less than 3°C. Further, the instantaneous nature of the response suggests that the larger portion of the response can be explained mechanically.

By displacing chromatin in a model of nuclear inversion (*MacPherson et al., 2020*; *Solovei et al., 2009*), we observed high rigidity of the centrally formed heterochromatin cluster despite large displacements. This provided further evidence that highly compacted heterochromatin does not behave like a liquid but rather like a solid at micron scale, as recently suggested (*Strickfaden et al., 2020*). More general, in conventional interphase nuclei, we found that chromatin motion was largely reversible and showed exponential asymptotic trends over time. The simplest model that fits the deformation and relaxation dynamics was a Kelvin-Voigt model that features a viscous dash pot and an elastic spring element connected in parallel, indicating that the underlying mechanisms of liquid and solid behavior are interwoven, in line with passive micro-rheology measurements that indicate fluid and gel-like material properties for euchromatin and heterochromatin, respectively, in differentiated chromatin, with the two relaxation times of 2.3 s and 0.8 s (*Eshghi et al., 2021*). Interestingly, time scales around 10 s for the viscous component were described previously for mammalian and yeast nuclei (*Pajerowski et al., 2007*; *Schreiner et al., 2015*). The characteristic times $\tau$ extracted from our data were around 1 s. Since $\tau$ reflects the ratio of viscosity to elasticity ($\tau = \eta/E$), this suggested that liquid and solid contributions of chromatin, specifically the phases C1–C6, are in close balance on the mesoscale when assessed on the time scale of seconds.

A recent study challenges the view of interphase chromatin as a gel-like material, highlighting the fluidity of chromatin, by the observation that near-piconewton forces can move a genomic locus across the nucleus over a few minutes (*Keizer et al., 2022*), though they do not exclude the possibility of gel-like patches embedded in a structure with liquid properties at a larger scale nor the possibility that chromatin may be a weak gel.

We further showed that different nuclear compartments possess distinct material properties. Specifically, we found that the susceptibility to volumetric deformation (absolute hydrostatic strain) showed a non-linear relationship over chromatin compaction with medium dense chromatin being most compliant, and lightly and densely packed chromatin being most resistant to deformation. A similar non-linear relationship between hydrostatic strain and chromatin density has been reported in cardiomyocyte nuclei during spontaneous cell contractions (*Ghosh et al., 2019*). That lightly packed chromatin compartments show high resilience is still somewhat surprising and merits further investigation. One reason could be that persistent tethering of RNA to transcribed chromatin, important for the formation of transcriptional pockets, provides structural support (*Hilbert et al., 2021*).

Furthermore, we observed a striking mechanical identity of the nucleolus, which showed higher resistance to deformation than any chromatin compartment. The nucleolus is considered to be a membraneless liquid droplet. Our findings might help to better understand the underlying physics (e.g. its surface tension) that allow the nucleolus to maintain a stable form (*Caragine et al., 2018*; *Feric et al., 2016*). We frequently observed that the nucleolus resisted displacement altogether. A reason for that could be that perinucleolar chromatin further anchors nucleoli to the nuclear lamina, especially in 2D cultures where nuclei have a flat topology (*Ghosh et al., 2019*). In such cases we found that perinucleolar chromatin showed similar resistance to displacement, suggesting a tight association to the nucleolus. Nucleolus associated domains (NADs) are thought to anchor peri-nucleolar heterochromatin to the nucleolus (*Canat et al., 2020*), but more needs to be understood. It was recently suggested that interfacial forces with their non-specific nature could play a role in a number of interactions of membraneless organelles with other supramolecular structures (*Böddeker et al., 2022*).

Based on recent FRAP experiments that revealed that chromatin does not mix and recover, while chromatin scaffold proteins rapidly do, the authors suggested that interphase chromatin is akin to a porous hydrogel. Our results support this view, and complement the FRAP-based evidence by detailed analysis of the mechanical relaxation response after perturbation. Specifically, our method reveals a characteristic dynamic behavior of a porous gel-like phase with both dissipative and elastic contributions, the latter of which being responsible for deformations being predominantly reversible. When analyzing chromatin motion around static nucleoli, one can directly observe that chromatin shows 'network effects', meaning coherent motion of spatially extended gel-like heterochromatin domains.

These network effects become visible over length scales of up to 1.5 µm as a smooth gradient of velocities surrounding immobile nucleoli and are consistent with coherent motion and relaxation of genetic loci after displacement by magnetic forces (*Keizer et al., 2022*).

As different methods shed light onto different aspects of nuclear organization, combining our approach with other complementary methodologies will be useful to reach an integrated view of nuclear organization. For example, combining data from live perturbations with chromatin conformation capture methods (Hi-C) might be key to connect mechanical identities of compartments with their underlying sequence interactions (*Hildebrand and Dekker, 2020*). Specifically, our method could be used in conjunction with the recently developed liquid Hi-C approach that aims to disentangle the contribution of the chromatin backbone and non-covalent chromatin interactions for nuclear mechanics and organization (*Belaghzal et al., 2021*). Similarly, ChIP-seq approaches could be employed to further elucidate the roles of epigenetic modifications and chromatin-protein binding in shaping these interactions (*Huang et al., 2015*; *Jiang and Mortazavi, 2018*; *Mourad and Cuvier, 2015*).

We also showed that this method allows to study the material interfaces between compartments, such as chromatin and the nucleoli. Similar approaches could be used to study the interaction of chromatin with the nuclear lamina, especially to study diseases in which lamina dysfunctions cause aberrant nuclear organization, referred to as laminopathies (*Isermann and Lammerding, 2013*; *Köhler et al., 2020*; *Stiekema et al., 2020*). Of high interest would also be to study the transition of this lamina-interaction during mitosis to achieve a better understanding of the underlying mechanism of nuclear reformation (*Serra-Marques et al., 2020*).

## Out of equilibrium physico-chemical driving forces

While our method constitutes a reliable and well tunable way to induce chromatin motion in cell nuclei and study its relaxation behaviors, our perturbations bear further potential to gain insight into the physical chemistry that underlies temperature-dependent chromatin organization. Temperature-dependent changes in chromatin compaction have been reported after bulk cooling of live cell nuclei (*Fischl et al., 2020*), albeit on the time scale of hours. Equally, on shorter time scales, reversible changes in nuclear volume have been observed after homogenous temperature increases (ΔT=18°C) in isolated nuclei (*Chan et al., 2017*). Interestingly, the study found that the sign of volumetric change was dependent on ion valency, especially multivalent cations, hinting toward an electro-osmotically driven influx of water into these isolated nuclei. As the directed motion of chromatin within a cell nucleus as described by us occurs without such nuclear volume changes and at about 10-fold smaller temperature differences (2°C vs 18°C), it is likely driven by the temperature gradient.

Temperature and pressure are frequently related, and an increase in temperature in a closed volume can sometimes generate strong increase in pressure. However, the average temperature increase over the nucleus is small, around 2°C. With a thermal expansion coefficient of water of $3.5 \times 10^{-4}$/°C, the volumetric expansion of the nucleus will remain below 0.1% when laser heated during our perturbations. Given the flexible nature of cell membranes, this is unlikely to cause significant changes in pressure.

A wide range of physical phenomena is known that give rise to the motion of microscopic objects in temperature gradients. The movement of molecules along a temperature gradient (thermophoresis or Soret effect) is complex and subject of ongoing scientific debates. However, studies have shown that the movement of highly charged polymers, such as DNA and RNA, in aqueous solutions can be predicted over a large range of experimental parameters by the temperature gradient-induced emergence of local and global electric fields that link ionic thermophoresis to electrostatic energies and the Seebeck effect respectively (*Duhr and Braun, 2006*; *Reichl et al., 2014*).

Moreover, the observed actuation of chromatin could in parts also be driven by a temperature-dependent affinity of DNA to histone complexes, potentially leading to a decompaction of condensed chromatin with increasing temperature. Equally, a temperature-dependent hydrophilicity of chromatin, as it is known for certain polymers (*Quesada-Pérez et al., 2011*) and has technologically been exploited to elicit responses phenomenologically similar to the bending of bi-metallic strips (*Hippler et al., 2019*), could potentially give rise to chromatin motion in temperature gradients. A recent study links the dynamic nature of nucleosomes to thermal fluctuations via molecular dynamics simulations (*Farr et al., 2021*). Additionally, motion in fluids may also be the result of the so-called thermoviscous

flows (*Weinert et al., 2008*). These have successfully been used to stream the cytoplasm (*Mittasch et al., 2018*), but require the spatial scanning of a temperature field, and as such can be decoupled from the here observed effects due to their fundamentally different symmetry properties of the stimulus.

In conclusion, we showed that strongly localized temperature gradients offer unexpected opportunities to study the organization of the living nucleus in a spatially resolved and dynamic manner.

## Materials and methods

### Cell culture and transfection

NIH-3T3 cells (ATCC, CRL-1658) were cultured in DMEM+GlutaMAX (Gibco) containing 10% fetal bovine serum (Gibco) and 1% penicillin-streptomycin (Gibco) at 37°C and 5% $CO_2$. The cell line was tested as negative for mycoplasma contamination. For temperature stimulation experiments, 150 μm thick c-axis cut sapphire cover slips (UQG Optics) were coated with fibronectin (60 μg/mm²) for 1 hr at RT and seeded with cells to reach 50% confluency the next day. Sapphire was chosen for its excellent heat conductivity while still allowing for high-quality imaging. Transfection of GFP-H2b, mCherry-H2b, or Casz1_v2 (NM_017766) containing plasmids was performed 18 hr after seeding using Lipofectamine 3000 and cells were incubated for another 24 hr before experiments. To visualize nucleoli, cells were stained with Cytopainter Nucleolar Staining Kit (Abcam, ab139475) 30 min before experiments. During experiments, a temperature chamber, consisting of a thick sapphire glass slide with Peltier elements on each side (*Mittasch et al., 2018*), was used to maintain a constant ambient temperature of 36°C (Peltier elements convert heat into energy and vice versa). On the day of experiments, sapphire cover slips containing transfected cells were mounted onto temperature chambers using 15 μm polystyrene spacer beads (Bangs Laboratories).

### Live cell imaging and temperature stimulation

Image stacks were taken on an inverted Olympus IX81 microscope equipped with a Yokogawa spinning disk confocal head (CSU-X1), 60×1.2 NA plan apochromat water objective and an iXon EM+DU-897 BV back illuminated EMCCD (Andor). Images were acquired at 4 frames per second, with an excitation of about 200 ms, using VisiView software (Visitron Systems). Cells were imaged for 1 min total, starting with 20 s of no stimulation (baseline), followed by 10 s of temperature stimulation and ending with 30 s of no stimulation again (reversibility).

To apply a local, precisely controlled temperature gradient, an infrared laser (1455 nm) was scanned along a line next to the nucleus at 1 kHz. The exact setup has been described before (*Mittasch et al., 2018*). Briefly, an infrared Raman laser beam (CRFL-20-1455-OM1, 20 W, near TEM00 mode profile, Keopsys) was acousto-optically scanned along a line. Precise deflection patterns were generated using a dual-channel frequency generator PCI card (DVE 120, IntraAction), controlled via modified LabVIEW (National Instruments) based control software (DVE 120 control, IntraAction), in combination with a power amplifier (DPA-504D, IntraAction). For two-dimensional laser scans (*Figure 4a*), a two-axis acousto-optical deflector (AA.DTSXY-A6-145, Pegasus Optik) was used. Precise laser scan patterns were performed by generating analog signals using self-written software in LabVIEW, in combination with a PCI express card (PCIe 6369, National Instruments). A dichroic mirror (F73-705, AHF, Germany) was used to couple the infrared laser beam into the light path of the microscope by selectively reflecting the infrared light but transmitting visible wavelengths which were used for fluorescence imaging.

### Dye-based temperature measurements

To measure the spatial temperature profile inside the cell incubation chamber, as well as the temperature increase inside nuclei during temperature stimulation experiments, we used temperature-sensitive decrease in quantum efficiency that has been well described for certain dyes (usually in the red spectrum) before (*Hirsch et al., 2018*; *Mittasch et al., 2018*; *Singhal and Shaham, 2017*). For chamber measurements, rhodamine B solution (Sigma, 02558) was diluted to 10% in water and image stacks were acquired in the red channel during laser application. For nuclear measurements, mCherry-H2b transfected cells were recorded. Both dyes were calibrated by precise changing of the bulk temperature of the incubation chamber (*Figure 1—figure supplement 1*). For rhodamine, thermophoretic

effects (lower dye intensity due to concentration difference, not quantum yield) were determined to correct measurements.

## Displacement and strain map calculation

A custom MATLAB code (*Source code 1*) was written to calculate spatial displacements and strain maps from image stacks of the fluorescence. PIV, in particular, a modified version of MatPIV (v 1.7) (*Sveen, 2004*), was used to generate displacement maps with a window size of 32 pixels, 75% overlap using multiple passes as well as local and global filters. The final displacement field resolution was 4×4 pixels and displacement maps in this manuscript are shown only at half resolution. From displacement maps, hydrostatic and shear strain maps were calculated according to:

$$\varepsilon_{\text{hydro}} = \frac{dudx + dvdy}{2 \cdot L_{\text{char}}}$$

$$\varepsilon_{\text{shear}} = \frac{dudy + dvdx}{L_{\text{char}}}$$

with $L_{\text{char}}$ being the characteristic (initial) length before deformation.

Hydrostatic strain is equal in all normal directions, with no shear components. It is a change in the volume of a body, but not its shape. Shear strain, on the other hand, is the ratio of change in dimensions to the original dimension due to shear stress and deformation perpendicular rather than parallel to it.

To extract local strain information, binary masks of chromatin densities, using intensity histograms, or of nucleoli regions using intensity thresholding were generated. Strain maps were extrapolated to match image resolution and local strains were averaged using binary masks. The same algorithm was used to track changes in nuclear and CHC area.

Image difference stacks and average image difference tracks were generated using ImageJ (v. 1.52t). The 'Analyze Particles' function in ImageJ was used to track the centroid position of nuclear features and calculate feature displacements. Dynamic measurements of image differences and displacements were fitted to a Kelvin-Voigt model, consisting of a spring and a dashpot in series, using the equations:

$$\varepsilon\left(t\right) = \frac{1}{E} \cdot \left(1 - e^{-t/\tau_c}\right)$$

$$\varepsilon\left(t > t_1\right) = \varepsilon\left(t_1\right) \cdot e^{-t/\tau_r}$$

$$\tau = \frac{\eta}{E}$$

For the creep (during stimulation) and relaxation phase (after stimulation), respectively, with E being the relative spring constant and $\tau$ the characteristic time, which reflects the ratio of viscosity ($\eta$) to elasticity (E) (*Meyers and Chawla, 2008*).

## Statistics

t-Tests or one-way ANOVA were conducted using OriginPro 2021 (v. 9.8.0.200). The statistical test used, as well as the number of repeats n and significance levels, are indicated in the figures and/or in the figure captions. All data with repeated measurements were collected in at least three independent experiments.

## Acknowledgements

We want to thank Falk Elsner and Claudius George for constructing the temperature incubation chambers. Further, we would like to thank Irina Solovei, Job Dekker, and Denis Lafontaine for discussions as well as Lennart Hilbert and Iain Patten for feedback on the manuscript, and Anatol Fritsch and Matthäus Mittasch for devising the setup used for the thermal manipulations. We acknowledge the European Research Council grant 'GHOSTs', #853619, for co-founding MK, SW and VRK, as well as a Life grant by the Volkswagen foundation for co-funding SW and IDS, and funding by the Max Planck Society. Further, we acknowledge funding by the German Research Foundation, the Hector Foundation and support of the Karlsruhe School of Optics & Photonics (financed by the Ministry of Science,

Research and the Arts of Baden-Wurttemberg as part of the sustainability financing of the projects of the Excellence Initiative II). We further acknowledge for funding the DFG grant (German Research Foundation) under Germany's Excellence Strategy, 2082/1—390761711, 3D Matter Made to Order, 3DMM2O.

## Additional information

### Competing interests

Elena Erben, Iliya Dimitrov Stoev: are listed as inventors of past patent applications that describe technology to stimulate biological samples with infrared light. Moritz Kreysing: listed as an inventor of past patent applicationsthat describe technology to stimulate biological samples with infrared light and further acts a consultant to Rapp Optoelektronik GmbH that commercializes related technologies. (JP2021504748A, EP3714308A1, US20200379235A1, WO2019101964A1, CN111373302A, EP4112171A1,WO2023274566A1, WO2023274565A1, WO2008077630A1). The other authors declare that no competing interests exist.

### Funding

| Funder | Grant reference number | Author |
| --- | --- | --- |
| European Research Council | 853619 | Susan Wagner<br>Susan Wagner |
| Volkswagen Foundation | Life grant 92772 | Iliya Dimitrov Stoev |
| Max Planck Institute of Molecular Cell Biology and Genetics | open access funding | Manavi Jain<br>Manavi Jain<br>Manavi Jain |
| Deutsche Forschungsgemeinschaft | 515462906 | Iliya Dimitrov Stoev |

The funders had no role in study design, data collection and interpretation, or the decision to submit the work for publication. Open access funding provided by Max Planck Society.

### Author contributions

Benjamin Seelbinder, Conceptualization, Software, Formal analysis, Investigation, Visualization, Methodology, Writing – original draft, Writing – review and editing, Interpretation of results; Susan Wagner, Manavi Jain, Formal analysis, Investigation, Writing – review and editing; Elena Erben, Investigation, Writing – review and editing; Sergei Klykov, Software, Methodology, Writing – review and editing; Iliya Dimitrov Stoev, Investigation, Writing – original draft, Writing – review and editing, Interpretation of results; Venkat Raghavan Krishnaswamy, Writing – review and editing; Moritz Kreysing, Conceptualization, Resources, Funding acquisition, Methodology, Writing – original draft, Writing – review and editing, Interpretation of results

### Author ORCIDs

Benjamin Seelbinder (ID) http://orcid.org/0000-0003-1004-4659
Susan Wagner (ID) http://orcid.org/0000-0002-7492-7541
Iliya Dimitrov Stoev (ID) https://orcid.org/0000-0003-3053-3548
Moritz Kreysing (ID) https://orcid.org/0000-0001-7432-3871

### Decision letter and Author response

Decision letter https://doi.org/10.7554/eLife.76421.sa1
Author response https://doi.org/10.7554/eLife.76421.sa2

## Additional files

### Supplementary files

• Transparent reporting form

• Source code 1. Custom MATLAB code to calculate spatial displacements and strain maps from image stacks.

## Data availability
The recorded video data for Fig. 1, 2, 3 and 5 are provided as rich media files in the supplements.

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
