## [Editor Report]

Seelbinder et al. describe a valuable new method for perturbing chromatin in living cells by strongly localized temperature gradients. Solid analysis reveals that chromatin shows both elastic and viscous properties at the timescales of seconds, with heterochromatin showing solid-like properties. While some details of the nuclear response to local heating remain to be elucidated, the ability of the method to reveal local mechanics in vivo makes the approach likely to be of broad interest to both the cell biophysics and cell biology communities.

---

## [Decision Letter]

**Decision letter after peer review:**

Thank you for submitting your article "Non-invasive Chromatin Deformation and Measurement of Differential Mechanical Properties in the Nucleus" for consideration by *eLife*. Your article has been reviewed by 3 peer reviewers, one of whom is a member of our Board of Reviewing Editors, and the evaluation has been overseen by Jessica Tyler as the Senior Editor. The following individual involved in the review of your submission has agreed to reveal their identity: Kristian Franze (Reviewer #2).

Essential revisions:

1. Validating the method as non-invasive: While it is true that the nuclear envelope remains intact in contrast to techniques that inject probes into the nucleus, the heat itself presents mechanical stress. The heat will increase the local pressure in the nucleus and may likely cause also non-physiological changes to the genome, the nucleus and the cell as a whole, as well as induce a stress response. In earlier studies mechanical stress has been shown to induce changes in the cell phenotype as well as the genome's organization and rheology (Denais et al., Science, 2016, Irianto et al., Curr. Biol., 2017, Pfeifer et al., Mol. Biol. Cell., 2018, Shah et al., Curr. Biol. 2021, Caragine et al., Soft Matter, 2022). Such effects should be discussed and control experiments assessing the "invasiveness" of the heat perturbation are needed. Moreover, the authors need to address whether the DNA is broken either from direct effects of the temperature perturbation/IR laser or indirect effects of the temperature effect on either biological processes that could drive DNA breaks or mechanical effects that could arise from tension exerted on chromatin-nuclear architecture contacts (for example, where chromatin is attached to the lamina). This could be achieved using cell lines expressing DNA repair factors as fluorescence proteins while carrying out the temperature gradient experiment. If the approach cannot be more fully vetted as being non-invasive, then the word "noninvasive" in the title might be better replaced with "heat-induced" chromatin deformation.

2. The authors need to provide a more thorough explanation of the origin of the forces leading to the movement of chromatin. Does the gradient cause a fluid flow directed down the temperature gradient (briefly explain why), which carries chromatin with it, or is there a direct effect on chromatin? Moreover, laser heating is expected to produce known strains that the authors measure – therefore an estimate of stresses exerted on chromatin should be provided. In addition, what is the effect of porosity on the motion of different compartments (if motion is driven by the movement of water through pores)? Similarly, although the temperature gradient likely introduces convective flows inside the nucleus, convection was not considered. This could affect the measured strains and thus the interpretation of the results and therefore needs to be discussed and its contribution estimated.

3. What is the basis for the irreversibility of the chromatin response, which seems to perhaps differ between different chromatin regions? The underlying factors that underlie this need to be further explored. For example, what could account for the chromatin not returning to the initial state? Are there different time scales here – in other words, if the nucleus is left unperturbed for tens of minutes will it return to the initial state? Are there chromatin entanglements that might arise (curious if topoisomerases are playing a role in that case)? If the authors do a second round with the IR laser do they recapitulate just the reversible component or do they also see an increase in the irreversible component? It's particularly interesting that the heterochromatin domain shows full reversibility (Figure 3). Is it more pronounced in the shear versus hydrostatic response (Figure 4b)? If so what does this tell us?

4. In addition, the interpretation of the application of local heat in an active system like the cell nucleus is nontrivial as the change in temperature will alter rates (k_on_, k_off_) of local chemical reactions as well as active processes. Thus, a pure mechanical interpretation of the heat effect might not be capturing all these effects. These caveats of the method should be discussed and assessed.

5. The authors need to acknowledge the time scales of behaviors that can be revealed using the approach and how this influences their observations. For example, they observe the creep behavior on the 1 second timescale, which is an order of magnitude below observations of the behavior of whole nuclei (~15 seconds) for nuclei from mammalian to yeast that has been suggested to reflect chromatin flow.

6. The authors may be unable to make conclusions about the rigidity of subcellular structures based on the presented experiments. For example, in the case of nucleoli, the reduced deformation observed by the authors is likely a consequence of surface tension (as mentioned by the authors towards the end of the discussion), which is not necessarily the same as rigidity. The latter usually refers to high elastic stiffness, which nucleoli likely don't possess. Something very stiff, on the other hand, could easily be moved in a fluid environment if it's not immobilized somehow. Hence, unless the authors can validate these statements with an alternative direct measurement of the nucleoli's mechanical properties, it would be best to avoid terms like rigid(ity) or stiff(ness).

7. It is not clear how the authors identify static versus mobile nucleoli. This should be clarified. Static nucleoli were found to be resistant to deformations. Was the possibility ruled out that the IR laser induces trapping forces at the nucleus due to its different refractive index from the surrounding nucleoplasm? This may also lead to an apparent resistance of nucleoli to deformation if they are effectively held in place. This should be ruled out as it can affect the interpretation of the observations.

8. The potential effect of the local temperature increase on the mechanical behaviour of nuclear compartments should at least be briefly discussed, even if deltaT did not exceed ~4-6K.

9. Further description of the strains calculated is needed. The readership may be unfamiliar with hydrostatic strains and in particular the difference between total vs. absolute. Please explain to a mostly biology readership what the different strains mean. The inset in Figure 4b is not easily understandable without further explanation.

10. There are numerous studies important for the premise and interpretation of this study that need to be considered/cited. In the Introduction, when referring to computer simulations of the genome separation into compartments, MacPherson et al., PNAS, 2018 and MacPherson et al., Biophys. J., 2020 should be included along with the currently cited Falk et al., Nature, 2019. Also in the Introduction, when reviewing prior methodology to study mechanical properties of the nucleus, some of the seminal works are presently missing, such as for micropipette aspiration (e.g., Dahl et al., J. Cell Sci., 2004, Dahl et al., Biophys. J., 2005, Pajerowski et al., PNAS, 2007), micromanipulation techniques (e.g. Stephens et al., Mol. Biol. Cell, 2017, Stephens et al., Mol. Biol. Cell, 2018), non-invasive techniques using natural and artificial probes (e.g., Shin et al., Cell, 2018, Caragine et al., Phys. Rev. Lett., 2018, Lee et al., Nature Physics, 2021, Caragine et al., Soft Matter, 2022) and spatially resolved non-invasive microrheology techniques (Eshghi et al., Phys. Rev. Lett., 2021). There are several studies across models for nuclear mechanics that provide a critical context for this work that are not discussed or cited including the different time and length scale regimes for the influence of chromatin and the nuclear lamina as dissected by Stephens et al. in a series of publications including probing the mechanical contributions for hetero- versus euchromatin (e.g. Stephens et al., MBoC, 2017, 2018); 2) context for the timescales across models for nuclei that is relevant to the creep response of chromatin from the Discher (e.g. Pajerowski et al., PNAS, 2007) and King lab (e.g. Schreiner et al., Nat Commun, 2015) and also observations in the Stephens works on the timescale and lengthscales on which chromatin versus the nuclear lamina dictate the force response of the nucleus; 3) studies on the mechanical probing of heterochromatin and particularly HP1 (Narlikar, Stephens). Last, the authors should compare their results with recent spatially resolved non-invasive rheology measurements (Eshghi et al., Phys. Rev. Lett., 2021), where heterochromatin was shown to behave as Kelvin-Voigt solid and euchromatin as Maxwell fluid with relaxation times comparable to those presented by authors. Also – how do the results of the current study compare to findings made in Keizer et al., 2021 cited by the authors?

[Editors’ note: further revisions were suggested prior to acceptance, as described below.]

Thank you for resubmitting your work entitled "Probe-free Optical Chromatin Deformation and Measurement of Differential Mechanical Properties in the Nucleus" for further consideration by *eLife*. Your revised article has been evaluated by Kevin Struhl (Senior Editor) and a Reviewing Editor.

While the reviewers continue to find the approach that your team has developed to be of interest and potential future utility for the field, the revised manuscript was found to be non-responsive to central concerns raised in the first round of review. There was a consensus that substantive issues with the text of the written manuscript must be addressed in order for it to be suitable for publication in *eLife*.

The primary concerns are two-fold. First, the key issues raised in the review were rebutted in the response to reviewers without any changes to the manuscript itself; as indicated below, these issues should be addressed by edits to the text – caveats, for example, need to be stated and therefore transparent to the reader. Second, it was felt that the expected rigor for a new methodology was somewhat lacking. At a minimum, the technical clarifications requested should be incorporated into the revised manuscript and more quantitative estimates should be included in the text.

Specifically:

1. With regards to the possibility that pressure increases in the nucleus in response to the heat applied by the method, it was felt that the authors dismissed the point and did not edit the manuscript to address the potential caveats or to relate quantitative estimates to support the notion that a change in pressure does not occur if, as they assert, the change in temperature is too small to have an influence. If the authors cannot support this assertion with evidence, they should at a minimum quantitatively address the point, as they did in the rebuttal, in the text.

2. The authors continue to assert that the method is non-invasive, but the reviewers are not fully convinced of this view, and expect that the authors to be somewhat more circumspect on this in the manuscript text. For example, while half of the cells did not present with stress granules after the laser stimulus, the other half did show such a response (new Figure 1 Figure Supp. 1h and revised text) – although the authors consider this not to be a "substantial" effect, the reviewers disagreed. The experiment should be discussed at greater length in the manuscript and the quantitative results should be explained to the reader in order to provide them with the evidence for the assertion.

3. A key point in the rebuttal was the statement that "…we would like to emphasise that we do not think the laser itself nor the short temperature increase up to 39{degree sign}C exert a considerable stress or produce a considerable strain…It is the sharp temperature gradient, created by the scan pattern that we apply, that induces the chromatin displacement (similar to the gradient in an electric potential causing the migration of ions in electrophoresis, rather than the absolute magnitude of the potential)." In addition to including the author's perspective on this point in the text of the paper, not just the rebuttal, we also ask that the support for this statement is expanded in the manuscript. For example, a quantitative physical argument or experimental evidence for this point needs to be provided, particularly as it lies at the heart of the new method.

---

## [Author Response]

Essential revisions:1. Validating the method as non-invasive: While it is true that the nuclear envelope remains intact in contrast to techniques that inject probes into the nucleus, the heat itself presents mechanical stress. The heat will increase the local pressure in the nucleus and may likely cause also non-physiological changes to the genome, the nucleus and the cell as a whole, as well as induce a stress response. In earlier studies mechanical stress has been shown to induce changes in the cell phenotype as well as the genome's organization and rheology (Denais et al., Science, 2016, Irianto et al., Curr. Biol., 2017, Pfeifer et al., Mol. Biol. Cell., 2018, Shah et al., Curr. Biol. 2021, Caragine et al., Soft Matter, 2022). Such effects should be discussed and control experiments assessing the "invasiveness" of the heat perturbation are needed. Moreover, the authors need to address whether the DNA is broken either from direct effects of the temperature perturbation/IR laser or indirect effects of the temperature effect on either biological processes that could drive DNA breaks or mechanical effects that could arise from tension exerted on chromatin-nuclear architecture contacts (for example, where chromatin is attached to the lamina). This could be achieved using cell lines expressing DNA repair factors as fluorescence proteins while carrying out the temperature gradient experiment. If the approach cannot be more fully vetted as being non-invasive, then the word "noninvasive" in the title might be better replaced with "heat-induced" chromatin deformation.

These are valid and interesting points, that motivated us to do additional experiments, and with which we engage in great depth below.

First of all, regarding the word “non-invasiveness”, we meant to stress that in contrast to optical or magnetic tweezer measurements, our method does not require to bring a mechanical probe or fluid into the nucleus, which would require to open cell and nucleus and would then constitute a foreign body in the nucleoplasm. We think this interpretation does justice to the contemporary use of ‘non-invasive’ in the biomedical field. Here, the term ‘non-invasive’ seems to originate from and clearly prevails in the classification of diagnostic techniques:

“Diagnostic techniques that do not involve the puncturing of the skin or incision, or the introduction into the body of foreign objects or materials are known as non-invasive

procedures.” (source: Dorland's Illustrated Medical Dictionary (32nd ed. 2012). Elsevier. p. 955)

Nevertheless, we feel that “non-invasive” in the title was collectively understood by the reviewers as ‘physiological’ and it might indeed have carried this co-notation. And we agree, we didn’t provide a strong enough focus on the physiology to justify the use of a term that reliably seems to evoke such associations. To be on the safe side and to prevent readers could be misguided by the title we decided to stress the ‘non-invasiveness’ we had in mind by describing it, and we changed the title to:

“Probe-free Optical Chromatin Deformation and Measurement of Differential Mechanical Properties in the Nucleus”

While a different discussion, and a claim that we did not intend to make, we acknowledge that also the topic of a possible physiological character of our perturbations is interesting and relevant.

In the following we discuss concerns regarding the physiological character of our perturbations as they have been raised by the reviewers. While doing so we categorize hypothetical mechanisms that could impact nuclear physiology according to: pressure, mechanical stress and DNA breaks, temperature and cellular stress response. We provide additional experiments where we managed to perform these, and reduce claims where we were not able to collect more experimental evidence.

i) Pressure:

It is true that in physics, temperature and pressure are frequently related, and an increase in temperature in a closed volume can sometimes generate strong increase in pressure.

We would like to state however, that the average temperature increase over the nucleus is small, around 2 kelvins. With a thermal expansion coefficient of water of 3.5 x 10^-4^ / kelvin, the volumetric expansion of the nucleus will remain below 0.1% when laser heated during our perturbations. Given the flexible nature of cell membranes, this is unlikely to cause significant changes in pressure.

ii) Mechanical stress and DNA breaks:

The reviewers made some very good points here. We acknowledge that according to the literature mechanical stress on the nucleus seems to be a known driver of nuclear responses, including nuclear stiffening.

Four of the five by the reviewers suggested studies (Denais et al., Irianto et al., Pfeifer et al. and Shah et al.) investigated the nuclear envelop integrity after migration through confined spaces and possible consequences for DNA integrity. We cite these studies in the revised manuscript in the introduction. The perturbations applied in these studies were quite heavy as cells had to pass through constrictions as small as 2 to 5 μm, which led to substantial deformation of the nucleus and nuclear envelop. We would like to point out that, during our perturbations, no deformation of the nuclear envelop was visible and the nuclear area changed only marginally (~1%) (Figure 2d-e, Video 3), *i.e.*, around 100 nm (Figure 2f-g). Because strains reported here likely differ by orders of magnitude from those we induced, we believe that consequences for DNA integrity as observed in those studies do not automatically apply for our perturbations.

The latter study, Caragine et al., which we cite now in the introduction as a reference for non-invasive techniques to investigate nuclear material properties, investigated chromatin dynamics after injection of a volume of 100 μm^3^ into the nucleus, a droplet with a diameter of around 4.6 μm. After the application of such mechanical stresses, 40% of the nuclei did not change their chromatin properties, while 60% of the nuclei displayed chromatin stiffening. In our study, the temperature gradient induced chromatin displacements were considerably smaller with only hundreds of nanometres to up to 2 μm (Figure 2b-e), suggesting an even milder long-term response of the chromatin, if at all.

Nevertheless, we decided to complement the re-submission with new data. Specifically, we conducted multiple rounds of experiments on the same nucleus to see if the response changes.

In nuclei with centrally compacted heterochromatin, where such measurements can be done most easily, we observed a slight decrease (less than 10%) in the amplitude of the strain after the first perturbation. This indicates a slight stiffening of the chromatin surrounding the centrally compacted heterochromatin after the first perturbation. Such stiffening of the nucleus after mechanical stress is known (Stephens et al., 2017), however we observe relatively weak stiffening, and definitely not in excess of what is reported in the literature.

We have included this result in the revised manuscript as an additional panel of Figure 4—figure supplement 1. We reference the new figure in the fourth results paragraph.

Regarding DNA breaks:

The laser we use is an infrared laser with a wavelength of 1455 nm and has less than a third of the photon energy than the one usually used for GFP excitation. In addition, the laser beam has much lower peak intensities than routinely used in confocal microscopy, especially because the focal spot is around 10 times bigger, lowering the intensity by 100 fold. With respect to multiphoton absorption processes, this even reduces the likelihood by 100^n^ fold, and it is thought to require at least 4 eV per photon to ionize DNA (Fernando et al., 1998). A photon at 1455 nm has hv = 0.85 eV of energy. We therefore believe, photon induced DNA damage through the infrared laser that we are using to induce chromatin displacement is very unlikely.

Further we would like to mention again, that nucleus deformation, including the deformation of the nuclear envelop, is a normal process for example when epithelial cells migrate or muscle fibers contract (Kalukula et al., 2022).

The recent study by Keizer et al., published in Science, succeeded in actively manipulating a genomic locus in a living cell by magnetic forces. Nevertheless, they dragged a genomic locus over a distance of several micrometer and for tens of minutes. With our approach, local shears are likely orders of magnitude lower, as we introduce local shear deformation of only 10%.

While also Keizer et al. did not monitor the extend of DNA breaks during their perturbations, and while we expect a much lesser amount of DNA breaks, if any, during our perturbations, we acknowledge, that it would be interesting to understand to which extend DNA breaks are introduced. For our next study, we are in the process of producing cell lines suitable to investigate this interesting question, but it goes beyond what we managed in this revision, despite considerable time invested.

An additional hint for the mildness of the mechanical stress introduced by our perturbations is given by the behavior of the nucleoli. A study by Dahl et al. showed that nucleoli deform with swelling of the nucleus (Dahl et al., 2005), while during our perturbations nucleoli did not deform.

As side note: we had interesting discussions in the lab of how much fibroblasts in our body get regularly deformed, e.g., just because we are sitting on them. Without being able to put a definite number on it, if nuclei get as much deformed as the embedding cells and tissue, nuclear deformation are very like associated with strain well in excess of 10%, and typically for more than 10 seconds, sometime hours.

iii) Temperature:

During our experiments, the average temperature across the cell is around 36°C and the peak temperature for most of the perturbations we show is around 39°C. For mammalian cells, also temperatures in small excess of 37°C should not constitute a problem or only mild heat stress, as body temperatures up to 38°C are known to occur in mice already at ambient temperature from 30°C (Kaplan and Leveille, 1974), as shown further below (our response to reviewers’ comment number 4).

As another example, bovine embryonic development in vitro was unaffected by temperatures of up to 40°C, while deleterious effects were observable at a temperature of 41°C (Rivera and Hansen, 2001). More so, during our perturbations, the cells experience a temperature of 39°C for only a few seconds.

Moreover, our temperature variations are below the extend that is typical in experimental practice, e.g., it is rather common to expose cells to much more severe temperature variations when preparing tissue culture in Matrigel. Here, the common practice is to mix mammalian cells and gels below the gelling point, *i.e*., at 4°C. Gelling is then achieved by bringing cells back to 37°C, which constitutes a temperature increase that is more than an order of magnitude stronger than during our perturbations. This procedure is implicitly regraded as physiological.

To investigate, if our cells experience larger damage through the laser stimulus and become necrotic or initiate apoptosis, we used propidium iodide (PI), a fluorescent intercalating agent, which is membrane impermeable. Our typical application of the laser stimulus with medium power did not lead to uptake of π by the cell, suggesting that the cell membrane is intact. We had to increase laser power by roughly five times in order to induce damage to the membranes and observe π uptake by the cell and nucleus, which made us conclude that no severe damage occurs at the laser powers we used.

We have included this experiment in the revised manuscript as panel e) of Figure 1—figure supplement 1 and refer to it in the manuscript in the first results paragraph. Please find the revised Figure 1—figure supplement 1at the end of our response to comment 1.

In order to understand, if the deformation of the chromatin by our perturbation requires the peak temperature to be above 36°C, or if it is the temperature gradient itself that drives the movement, we performed our perturbations at 30°C. We observed the same response as performed at 36°C and conclude that the temperature gradient is responsible for the movement of the chromatin but not temperatures higher than 36°C as such. Therefore, a stress response or other unspecific responses to unphysiologically high temperature can also be excluded as driver for the chromatin displacement. We discuss a possible stress response of the cell to our perturbation in point v) below.

We have included this experiment as panels c) and d) of Figure 1—figure supplement 1, and refer to it in the manuscript in the first results paragraph.

iv) Stress response:

We decided to investigate if the temperature stimulus determines as stress response in treated cells. For this, we obtained HeLa cell lines with G3BP1 tagged with GFP or mCherry. G3BP1 is a well-known marker for stress granules. First of all, we repeated the original experiments also on these human cell lines and observed comparable chromatin displacements as previously in the mouse fibroblasts.

Next, we asked whether cells form stress granules after the application of the laser stimulus. A HeLa cell line stably expressing the stress granule marker gene G3BP1 with a C-terminal fusion to mCherry showed varying formation of stress granules after the laser stimulus. Treatment with Thapsigargin, a chemical known to introduce stress granules, lead to the expected formation of stress granules without laser stimulus. After application of our laser stimulus, in seven repetitions we observed either no stress granule formation (three times), the formation of a few stress granules (two times) or strong stress granule formation (two times) within 15 minutes after a typical laser stimulus of 10 seconds. Similar behavior was found in HeLa cells stably expressing G3BP1 fused to GFP.

We concluded, that there is no obligatory stress granule formation after using laser heating at 36°C, *i.e.*, the laser stimulus alone is not sufficient to necessarily cause a stress response, but, depending on the state of the cells the additional stimulus of the laser can lead to the formation of a stress response. Due to the mounting of the cells inside the thin chambers, they experience other stresses, pre-stresses, and the laser heating in addition might trigger the actual stress granule formation. This is then likely due to the short temperature increase above 36°C. Yet, in near half of the cases, our perturbations alone did not trigger any stress granule formation at all.

We have included three additional panels to Figure 1—figure supplement 1, panels f) to h), to present our results about stress granule formation after laser stimulus. We refer to it in the revised manuscript with a short statement about our findings in the first result paragraph:

‘…without triggering a substantial stress response (Figure 1—figure supplement 1f-h).’

Last but not least, we feel it is fair to say that while the points raised are insightful and interesting, we are not aware of classic probe dependent micro-rheology studies that managed to look at such a wide range of down-stream consequences for cell fate, stress response, heat shock, pressure increase.

2. The authors need to provide a more thorough explanation of the origin of the forces leading to the movement of chromatin. Does the gradient cause a fluid flow directed down the temperature gradient (briefly explain why), which carries chromatin with it, or is there a direct effect on chromatin? Moreover, laser heating is expected to produce known strains that the authors measure – therefore an estimate of stresses exerted on chromatin should be provided. In addition, what is the effect of porosity on the motion of different compartments (if motion is driven by the movement of water through pores)? Similarly, although the temperature gradient likely introduces convective flows inside the nucleus, convection was not considered. This could affect the measured strains and thus the interpretation of the results and therefore needs to be discussed and its contribution estimated.

Firstly, we would like to answer the comment about the laser heating being expected to produce known strains and what would be our estimate of the exerted stresses. Here, we would like to emphasise that we do not think the laser itself nor the short temperature increase up to 39°C exert a considerable stress or produce a considerable strain as we have outlined in our response to comment 1 (please refer to section ii) Mechanical stress and DNA breaks). It is the sharp temperature gradient, created by the scan pattern that we apply, that induces the chromatin displacement (similar to the gradient in an electric potential causing the migration of ions in electrophoresis, rather than the absolute magnitude of the potential).

Regarding a thorough explanation, an exact physical explanation for such a phenomenon may not even exist up to today. As part of the extensive discussion of our manuscript, we had discussed the physical effects that may or may not contribute to the deformation of chromatin in the cell nucleus in detail in the last three paragraphs of the discussion starting on page 15: “While our method…”. But, we acknowledge that we could have done a better job in labeling this section explicitly, which we now did.

The subheading ‘Out of equilibrium physico-chemical driving forces’ was introduced.

To follow up on the reviewer’s suggestion of fluid and convective flows, one in our manuscript discussed possibility are fluid flows induced by the gradient in temperature and viscosity, so called thermoviscous flows. Such flows are unrelated to the convective momentum current. We excluded thermoviscous flows as an explanation of our observations due to the fundamentally different symmetry properties of the stimulus. Therefore, we do not expect fluid streaming through the nuclear pores as a primary cause for the chromatin displacment, as suggested by the reviewers.

To give a more comprehensive summary of our discussion, time dependent and especially inhomogeneous temperature stimuli, such as the temperature gradient we induce, are known to drive a wide range of physical phenomena on the micron-scale. We explain that it is likely the temperature gradient that drives the chromatin displacement and we discuss four physical phenomena that might give rise to the motion of microscopic objects in temperature gradients: i) Temperature gradient-induced emergence of local and global electric fields by which movement of highly charged polymers, such as DNA and RNA, in aqueous solutions can be predicted. ii) Temperature dependent affinity of DNA to histone complexes. iii) Temperature dependent hydrophilicity of chromatin. iv) Thermoviscous flows.

We extended our discussion in the revised manuscript by a recent study that links the dynamic nature of nucleosomes to thermal fluctuations via molecular dynamics simulations (Farr et al., 2021, Nature).

And while it might seem unsatisfying that we cannot calibrate the ‘forces’ based on our method alone, we wish to add that:

i) Non-equilibrium thermodynamics is a field that is currently not fully developed and is not conceptually understood,

ii) and it is not clear if there are any forces involved at all.

Thermophoresis has long been used but the effect is not fully described despite continuing debates about the microscopic origins of the effect.

3. What is the basis for the irreversibility of the chromatin response, which seems to perhaps differ between different chromatin regions? The underlying factors that underlie this need to be further explored. For example, what could account for the chromatin not returning to the initial state? Are there different time scales here – in other words, if the nucleus is left unperturbed for tens of minutes will it return to the initial state? Are there chromatin entanglements that might arise (curious if topoisomerases are playing a role in that case)? If the authors do a second round with the IR laser do they recapitulate just the reversible component or do they also see an increase in the irreversible component? It's particularly interesting that the heterochromatin domain shows full reversibility (Figure 3). Is it more pronounced in the shear versus hydrostatic response (Figure 4b)? If so what does this tell us?

We are glad to see that our method stimulated a wide range of ideas for follow up experiments to gain insight into numerous aspects of nuclear organization.

The displacement of the chromatin that we observed was mainly reversible. Nevertheless, we have seen in our analysis that there are residual differences after the perturbation, irreversible, and well fitted with a partial exponential decay of strain fields after perturbation. Irreversible changes on time scales of tens of seconds and beyond are known to happen spontaneously, even without external perturbations. These specifically include the coherent motion of chromatin domains on the same length scale that we investigated (Zidovska et al., 2013, PNAS). We also did observe some chromatin movement without laser stimulus (Figure 1f, control) that likely reflects the spontaneous coherent motion of chromatin and might account for some of the residual differences after perturbations. To assess if the changes fully reverse on longer time scales, as proposed by the reviewers would be challenging because the signal would get lost in the noise of spontaneous chromatin movement.

Noteworthy, pounced (ca 50%) irreversible displacements can also be observed when pulling a genomic locus via a magnetic probe through the nucleus albeit at approximately 10 times slower scales. (Figure 1, Keizer et al. 2022 Science). The authors fitted a Rouse polymer model of DNA inside the nucleus which predicted a recoil proportional to the total displacement during the pull. However, in many cases, they observed recoils being more slowly than predicted and those deviations were more pronounced at the nuclear periphery. The authors did not discuss any theories about the basis of the irreversibility.

We have updated the reference Keizer et al. in the revised manuscript, as the preprint got published in the journal Science in the meantime.

The question if different strain fields decay on different time scales is also an interesting one. At least for the type of experiments that we did so far, we observed that indeed the time scales seem to match, as indicated in Figure 2e.

As the reviewers mentioned, it is also interesting to see that different chromatin phases are differently affected. Here one should distinguish the motion of some domains from their deformation. We have seen that highly compacted heterochromatin acts largely as a solid, which moves but does not deform. Our interpretation of the higher degree of reversible motion is that large solid heterochromatin domains are sufficiently anchored to the lamina via elastic tethers to enable reversible motion, while more fluid like phases of less compacted chromatin possess many internal degrees of freedom that allow to dissipate externally acting forces, leading to irreversible motion.

The reviewers were also asking about the distinction of the shear and hydrostatic strain, that we present in Figure 4b. There is a difference in magnitude, but both basically return to their original value. There is no substantial difference in the reversibility of shear and hydrostatic strains.

Last but not least, we have performed repeated rounds of laser stimulations on cells with centrally compacted heterochromatin and observed a slight stiffening. That means the magnitude of displacement slightly decreased after the first laser stimulus. We have discussed these results in our response to the reviewers’ comment no. 1 in section ii) Mechanical stress and DNA breaks. We have included this result into the revised manuscript as Figure 4—figure supplement 1b.

4. In addition, the interpretation of the application of local heat in an active system like the cell nucleus is nontrivial as the change in temperature will alter rates (k_on_, k_off_) of local chemical reactions as well as active processes. Thus, a pure mechanical interpretation of the heat effect might not be capturing all these effects. These caveats of the method should be discussed and assessed.

Though, our perturbations are very short in time and the induced temperature gradient is within only 3°C, we cannot exclude that these perturbations have on effect on the rate of local chemical reactions. Nevertheless, owing to time scales of only a few seconds, modifications of reaction rates will to only marginal change of local concentrations of products and educts. Further, the instantaneous nature of the response to the strain suggests that the larger portion of the response can be explained mechanically.

While we don’t know for sure if this is what the reviewers had in mind, we feel they might be concerned about a chemical dis-equilibrium akin to the trafficking of goods across the border of two neighboring states with individually functional but different tax systems. This is an interesting thought, albeit we do not see strong evidence for it.

It might be that differential temperatures modulate chromatin compaction differently. How this is achieved is an interesting question, and we think a sufficient mechanism would be the temperature depended swelling of hydrophilic polymers, as discussed above, in our answer addressing comment 2.

To express the uncertainty about possible chemical effects we have included the following sentences into the first discussion paragraph of the revised manuscript:

“Although, we cannot exclude that rates for biochemical reactions within the cell might be altered moderately due to the change in temperature, the change of local concentrations of products and educts will be marginal due to time scales of only few seconds and the narrow range of temperature gradient of less than 3°C. Further, the instantaneous nature of the response suggests that the larger portion of the response can be explained mechanically.”

Additionally, we would like to note examples showing that the range of temperature fluctuations that naturally occur within tissues and cells of organisms or that cells experience during common experimental practices are much wider than commonly assumed.

Cells are made to experience strong changes in temperature. Even in humans with comparably low surface to volume ratio, parts of our body, *e.g.*, fingers, are known be down regulated in temperature to less than 20°C within approximately 10 minutes of work in moderately cold environments (Ceron et al., 1995). And while temperature will slightly vary from the inside of the finger to the surface, fibroblast in these tissues will experience the same temperatures.

This being said, physiological temperature variations of cells inside the body are likely 10 times as big as the ones that we induce. Yet, working under standardized conditions, temperature variations that are tolerated by cells, might still influence the cells.

This raises the question how well temperature is typically controlled in microscopy?

The arguable most temperature sensitive cells are fertilized human egg cells, and considerable efforts are made to ensure highest temperature stability. This is reflected in the elevated prices of IVF hoods, incubators and microscopes and the dedicated personal training.

When independently assessed and measured with third party professional equipment, a study by UK and US based research, comparing temperature conditions in hood and on surfaces of 36 reproduction clinics found that even for IVF work certified equipment, displayed temperatures differed significantly from actually measured temperatures. The researchers found that the average of differences between the displayed and measured mean temperatures were as big as 1.36°C and 1.32°C for hood surfaces and microscope surfaces, respectively (Palmer et al., 2019).

And while we did our best to keep the ambient temperature of our sample constant, we wish to say that the temperature perturbations that we induce with the laser would be for the most part lost in the temperature fluctuations of some of the best temperature controlled biomedical scopes, as the ones for IVF work.

In conclusion, the temperature variations that we induce in the sample are not bigger than temperature fluctuations that even occur in the best controlled scopes for bio-medical use for much more fragile samples, and they are even 10 times smaller than the temperature variation that fibroblast routinely cope with in our bodies.

As a last point, we would like to mention that body temperature of mice is not really constant but may vary by up to 5 degrees depending on ambient temperature, (Kaplan and Leveille, 1974), and it is simply not correct that only 37°C should be seen as physiological temperature for mouse cells.

5. The authors need to acknowledge the time scales of behaviors that can be revealed using the approach and how this influences their observations. For example, they observe the creep behavior on the 1 second timescale, which is an order of magnitude below observations of the behavior of whole nuclei (~15 seconds) for nuclei from mammalian to yeast that has been suggested to reflect chromatin flow.

It is an interesting question why different experiments may show some differences in mechanical responses. To start with, we think that the choice of the cell line might be of some importance. Moreover, we think that it may indeed matter if nuclei are probed locally or fully. The deformation inside the nucleus might partially be subjected to poroelastic friction terms, that are known to elicit length scale dependent relaxation dynamics.

*I.e.*, a sponge that re-fills with water after deflation might do that on a different time scale than the elastic recoil that it locally shows after having pulled locally on its surface with a tweezer. This might also reflect that on different length scale different physical effect might play a role.

Response times around 1s, similar to what we observed, have also been reported in the literature for the nuclear region of MCF7 cells using an atomic force microscope (Moreno-Flores et al., 2010) and by Eshgi et al., 2021, for differentiated chromatin of embryonic stem cells.

We have extended our discussion on timescales in the revised manuscript to provide the reader with a wider context. Please see our response to the reviewers’ comment number 10 (part v).

6. The authors may be unable to make conclusions about the rigidity of subcellular structures based on the presented experiments. For example, in the case of nucleoli, the reduced deformation observed by the authors is likely a consequence of surface tension (as mentioned by the authors towards the end of the discussion), which is not necessarily the same as rigidity. The latter usually refers to high elastic stiffness, which nucleoli likely don't possess. Something very stiff, on the other hand, could easily be moved in a fluid environment if it's not immobilized somehow. Hence, unless the authors can validate these statements with an alternative direct measurement of the nucleoli's mechanical properties, it would be best to avoid terms like rigid(ity) or stiff(ness).

These are valid reflections. We agree that rigidity is often associated with elastic stiffness. And while also the deformation of a round droplet that has its shape due to surface tension, is strictly speaking itself an elastic (meaning energy conserving) deformation, we understand the concerns of the reviewers and we replaced ‘rigidity’ with ‘resistance’ when describing the response of nucleoli (Abstract).

7. It is not clear how the authors identify static versus mobile nucleoli. This should be clarified. Static nucleoli were found to be resistant to deformations. Was the possibility ruled out that the IR laser induces trapping forces at the nucleus due to its different refractive index from the surrounding nucleoplasm? This may also lead to an apparent resistance of nucleoli to deformation if they are effectively held in place. This should be ruled out as it can affect the interpretation of the observations.

We generally observed chromatin motion away from the temperature stimuli, down the temperature gradients, while the laser was always carefully positioned next to the cell nucleus, which caused the unidirectional motion of chromatin along the temperature gradients. We had indicated the scan line in figure 5a, panel 2, but we forgot to mention this in the caption. In the revised manuscript the caption of figure 5a was updated accordingly. We have also added a note to Material and Methods that the scan line was placed next to the nucleus.

Optical trapping is an interesting idea to explain the static behavior of some nucleoli, but this would have required to put the laser scan path over the nucleolus, which we did not in our study. Nevertheless, we were interested and have tested placing the scan line across the nucleus. This yielded a fundamentally different response. We did not observe unidirectional movement, but movement of chromatin to both sides away from the scan line, rather than towards it, as one would expect for optical trapping effects (Author response image 1).

**Author response image 1. sa2fig1:** Chromatin displacement to both sides away from the laser scan line is induced when the scan line is placed across the nucleus. Response of NIH-3T3 nuclei to an applied temperature gradient at medium laser intensity, performed at an ambient temperature of 36°C. As routinely performed and described in Figure 1 and Materials and methods, with the difference that cells were not transfected with labeled H2b, but stained with Hoechst and that the scan line (red dotted line) was placed across the nucleus instead of next to the nucleus. An overlay of the displacement map and chromatin density is shown.

8. The potential effect of the local temperature increase on the mechanical behaviour of nuclear compartments should at least be briefly discussed, even if deltaT did not exceed ~4-6K.

Throughout our perturbations we used a medium laser power which leads to a local temperature increase smaller than 3.5 K in the nucleus (Figure 1c, red line). The δ T the reviewers mentioned, ~ 4 to 6K, is those for high laser power which was not used.

Generally, we refer to our response to comments 1 and 4, regarding the physiological temperature range of the cell in nature, and the temperature fluctuations in the best controlled biomedical microscopes, which are on the order of the temperatures that we induce.

We also would like to confirm that it is the gradient of temperature but not absolute temperatures above 36°C that induce the chromatin displacement, we have performed our perturbations at 30°C ambient temperature which led to the same chromatin displacement that we observed at 36°C ambient temperature. We have included those results in the revised manuscript as Figure 1—figure supplement 1d (see our response to comment 1).

Therefore, we strongly believe that the induced temperature increase does not change the mechanical behavior of nuclear components drastically, if at all, and that infrared laser induced temperature gradients are an insightful tool to study mechanics of cell compartments.

9. Further description of the strains calculated is needed. The readership may be unfamiliar with hydrostatic strains and in particular the difference between total vs. absolute. Please explain to a mostly biology readership what the different strains mean. The inset in Figure 4b is not easily understandable without further explanation.

In our original manuscript we provided the following explanation of hydrostatic and shear strains in the fourth result chapter: ‘Shown here are local volumetric changes (hydrostatic strain) and orthogonal displacements (shear strain).’ A further explanation has been included in the revised manuscript to the corresponding section in Materials and methods: ‘Hydrostatic strain is equal in all normal directions, with no shear components. It is a change in the volume of a body, but not its shape. Shear strain, on the other hand, is the ratio of change in dimensions to the original dimension due to shear stress and deformation perpendicular rather than parallel to it.’

We also gave a short explanation of total vs. absolute strain in the fourth result chapter:

“Averaging non-absolute (total) hydrostatic strains over the whole nucleus, where positive values (extension) and negative values (compression) can cancel each other”

Additionally, we have included into the revised manuscript a description of absolute and total strain to the caption of the corresponding figure panels, 4d and 4e, respectively:

“d) Magnitudes of absolute hydrostatic strains (the sign of the strain is not considered, meaning positive values (extension) and negative values (compression) are added up)”

and

“e) Local analysis of averaged total hydrostatic strains (the sign of the strain is considered meaning positive values (extension) and negative values (compression) cancel each other)”

10. There are numerous studies important for the premise and interpretation of this study that need to be considered/cited. In the Introduction, when referring to computer simulations of the genome separation into compartments, MacPherson et al., PNAS, 2018 and MacPherson et al., Biophys. J., 2020 should be included along with the currently cited Falk et al., Nature, 2019. Also in the Introduction, when reviewing prior methodology to study mechanical properties of the nucleus, some of the seminal works are presently missing, such as for micropipette aspiration (e.g., Dahl et al., J. Cell Sci., 2004, Dahl et al., Biophys. J., 2005, Pajerowski et al., PNAS, 2007), micromanipulation techniques (e.g. Stephens et al., Mol. Biol. Cell, 2017, Stephens et al., Mol. Biol. Cell, 2018), non-invasive techniques using natural and artificial probes (e.g., Shin et al., Cell, 2018, Caragine et al., Phys. Rev. Lett., 2018, Lee et al., Nature Physics, 2021, Caragine et al., Soft Matter, 2022) and spatially resolved non-invasive microrheology techniques (Eshghi et al., Phys. Rev. Lett., 2021). There are several studies across models for nuclear mechanics that provide a critical context for this work that are not discussed or cited including the different time and length scale regimes for the influence of chromatin and the nuclear lamina as dissected by Stephens et al. in a series of publications including probing the mechanical contributions for hetero- versus euchromatin (e.g. Stephens et al., MBoC, 2017, 2018); 2) context for the timescales across models for nuclei that is relevant to the creep response of chromatin from the Discher (e.g. Pajerowski et al., PNAS, 2007) and King lab (e.g. Schreiner et al., Nat Commun, 2015) and also observations in the Stephens works on the timescale and lengthscales on which chromatin versus the nuclear lamina dictate the force response of the nucleus; 3) studies on the mechanical probing of heterochromatin and particularly HP1 (Narlikar, Stephens). Last, the authors should compare their results with recent spatially resolved non-invasive rheology measurements (Eshghi et al., Phys. Rev. Lett., 2021), where heterochromatin was shown to behave as Kelvin-Voigt solid and euchromatin as Maxwell fluid with relaxation times comparable to those presented by authors. Also – how do the results of the current study compare to findings made in Keizer et al., 2021 cited by the authors?

We thank the reviewers for the rich and important input.

All suggested publications have been included into the revised manuscript.

i) To the introduction as reference for various methods to study mechanical properties of the nucleus the following studies were added: MacPherson et al., 2018, 2020; Eshghi et al., 2021; Dahl et al., 2004, 2005; Pajerowski et al., 2007; Stephens et al., 2017, 2018; Caragine et al., 2018, 2022; Lee et al., 2021; Shin et al., 2018

ii) The Work by Stephens et al. 2017 and two works of the Narlikar lab are mentioned in the introduction: ‘…and increase in euchromatin leads to softening of the nucleus (Stephens et al., 2018). Recent studies demonstrated that chromatin compaction by HP1 proteins results in phase-separated liquid condensates (Sanulli et al., 2019; Keenen et al., 2021).’

iii) The Work by Stephens et al. 2018 is mentioned in the introduction: ‘…while chromatin governs response to small extensions (<3 μm) and the lamina to larger extensions (Stephens et al., 2017).’

iv) The Work by Keizer et al. 2022 is now more extensively discussed: ‘A recent study challenges the view of interphase chromatin as a gel-like material, highlighting the fluidity of chromatin, by the observation that near-piconewton forces can move a genomic locus across the nucleus over a few minutes (Keizer et al., 2022), though they do not exclude the possibility of gel-like patches embedded in a structure with liquid properties at a larger scale nor the possibility that chromatin may be a weak gel.’

v) To the discussion, we have added a reflection of the rheology measurements of differentiated chromatin by Eshghi et al. and have added a context for the time scales. It reads: ‘… in line with passive micro-rheology measurements that indicate fluid and gel like material properties for euchromatin and heterochromatin, respectively, in differentiated chromatin, with the two relaxation times of 2.3 s and 0.8 s (Eshghi et al., 2021). Interestingly, timescales around 10 s for the viscous component were described previously for mammalian and yeast nuclei (Pajerowski et al., 2007; Schreiner et al., 2015).’

References:

Ceron, R. J., Radwin, R. G., and Henderson, C. J. (1995). Hand Skin Temperature Variations for Work in Moderately Cold Environments and the Effectiveness of Periodic Rewarming. *American Industrial Hygiene Association Journal*, *56*(6), 558–567. https://doi.org/10.1080/15428119591016782

Dahl, K. N., Engler, A. J., Pajerowski, J. D., and Discher, D. E. (2005). Power-Law Rheology of Isolated Nuclei with Deformation Mapping of Nuclear Substructures. *Biophysical Journal*, *89*(4), 2855–2864. https://doi.org/10.1529/biophysj.105.062554

Fernando, H., Papadantonakis, G. A., Kim, N. S., and LeBreton, P. R. (1998). Conduction-band-edge ionization thresholds of DNA components in aqueous solution. *Proceedings of the National Academy of Sciences*, *95*(10), 5550–5555. https://doi.org/10.1073/pnas.95.10.5550

Kalukula, Y., Stephens, A. D., Lammerding, J., and Gabriele, S. (2022). Mechanics and functional consequences of nuclear deformations. *Nature Reviews Molecular Cell Biology*, *23*(9), 583–602. https://doi.org/10.1038/s41580-022-00480-z

Kaplan, M. L., and Leveille, G. A. (1974). Core temperature, 02 consumption, and early detection of ob/ob genotype in mice. *AMERICAN JOURNAL OF PHYSIOLOGY*, *227*(4), 912–915.

Moreno-Flores, S., Benitez, R., Vivanco, M. dM, and Toca-Herrera, J. L. (2010). Stress relaxation and creep on living cells with the atomic force microscope: a means to calculate elastic moduli and viscosities of cell components. *Nanotechnology*, *21*(44), 445101. https://doi.org/10.1088/0957-4484/21/44/445101

Palmer, G. A., Kratka, C., Szvetecz, S., Fiser, G., Fiser, S., Sanders, C., Tomkin, G., Szvetecz, M. A., and Cohen, J. (2019). Comparison of 36 assisted reproduction laboratories monitoring environmental conditions and instrument parameters using the same quality-control application. *Reproductive BioMedicine Online*, *39*(1), 63–74. https://doi.org/10.1016/j.rbmo.2019.03.204

Rivera, R. M., and Hansen, P. J. (2001). Development of cultured bovine embryos after exposure to high temperatures in the physiological range. *Reproduction*, *121*, 107–115.

Stephens, A. D., Banigan, E. J., Adam, S. A., Goldman, R. D., and Marko, J. F. (2017). Chromatin and lamin A determine two different mechanical response regimes of the cell nucleus. *Molecular Biology of the Cell*, *28*(14), 1984–1996. https://doi.org/10.1091/mbc.e16-09-0653

[Editors’ note: what follows is the authors’ response to the second round of review.]

The primary concerns are two-fold. First, the key issues raised in the review were rebutted in the response to reviewers without any changes to the manuscript itself; as indicated below, these issues should be addressed by edits to the text – caveats, for example, need to be stated and therefore transparent to the reader. Second, it was felt that the expected rigor for a new methodology was somewhat lacking. At a minimum, the technical clarifications requested should be incorporated into the revised manuscript and more quantitative estimates should be included in the text.Specifically:1. With regards to the possibility that pressure increases in the nucleus in response to the heat applied by the method, it was felt that the authors dismissed the point and did not edit the manuscript to address the potential caveats or to relate quantitative estimates to support the notion that a change in pressure does not occur if, as they assert, the change in temperature is too small to have an influence. If the authors cannot support this assertion with evidence, they should at a minimum quantitatively address the point, as they did in the rebuttal, in the text.

This is actually true, sorry for the oversight. As requested, we now included into the manuscript the points why pressure is unlikely to play a role in the process, which we previously included in the rebuttal letter. Specifically, the following paragraph was included into the second part of the discussion.

“Temperature and pressure are frequently related, and an increase in temperature in a closed volume can sometimes generate strong increase in pressure. However, the average temperature increase over the nucleus is small, around 2°C. With a thermal expansion coefficient of water of 3.5 x 10^-4^ /°C, the volumetric expansion of the nucleus will remain below 0.1% when laser heated during our perturbations. Given the flexible nature of cell membranes, this is unlikely to cause significant changes in pressure.”

2. The authors continue to assert that the method is non-invasive, but the reviewers are not fully convinced of this view, and expect that the authors to be somewhat more circumspect on this in the manuscript text. For example, while half of the cells did not present with stress granules after the laser stimulus, the other half did show such a response (new Figure 1 Figure Supp. 1h and revised text) – although the authors consider this not to be a "substantial" effect, the reviewers disagreed. The experiment should be discussed at greater length in the manuscript and the quantitative results should be explained to the reader in order to provide them with the evidence for the assertion.

We understand the need to explain more deeply our view on the extend of stress granule formation after the laser stimulus. We do that now in the first results paragraph. Although, we did not change our conclusion. The observation of zero stress granule formation in three out of seven repetitions shows that the stimulus alone is not sufficient to trigger a stress response (if the laser stimulus was sufficient to induce stress granule formation, it would happen every time). As the positive control clearly showed that the cells are stress-granule-competent, we conclude that apart from the laser stimulus other factors are required to explain why some cells showed stress repones and some not. As it is known that stress sensitivity can be increased by pre-stressing cells. The literature reports on the frequent and recent use of arsenate for this (*e.g.* Wheeler at al., 2019, bioRxiv, https://doi.org/10.1101/721001), but it seems equally established that also other forms of stress might be additive in nature, and we think such stressors cannot always be excluded in our experimental set up, *i.e.* when cells are mounted in a thin chamber of approximately 15 μm height, which might lead to transient pressures during the mounting process.

Therefore, the full experimental set up can lead to a stress response, although milder than the stress response triggered by thapsigargin treatment. But again, strictly speaking the laser stimulus alone does not seem to be sufficient to trigger a stress response.

The paragraph we have added to the first results paragraph reads as follows:

“The laser stimulus did not trigger a substantial stress response (Figure 1—figure supplement 1f-h). Formation of dynamic stress granules (SGs) is a typical and reversable response of cells to mitigate several kinds of stress (Hofmann et al., 2021). The protein G3BP1, for example, is a marker of SG. Under normal conditions, G3BP1 is distributed in the cytoplasm, but accumulates into granules when a cell experiences a stress (Hofmann et al., 2021), such as short treatment with thapsigargin (Sidrauski et al., 2015; Figure 1—figure supplement 1g). Compared to the SG formation upon treatment with thapsigargin, we observed only minor formation of SGs if at all (Figure 1—figure supplement 1h). Within 15 minutes after a typical laser stimulus of 10 seconds, in seven repetitions we observed either no SG formation (three times), the formation of a few SGs (two times) or stronger SG formation (two times), but still to a lower extend compared to the positive control of thapsigargin treatment (Figure 1—figure supplement 1h). We concluded, that the laser stimulus alone is not sufficient to trigger a stress response, although, additional stresses caused by the experimental set up, for example, mounting of the cells inside the thin chambers, can in sum lead to SG formation in some cases. Yet, in near half of the cases, our perturbations did not trigger any stress granule formation. This indicates that the laser induced chromatin motion does not necessarily evoke a stress response. However, some stress response was observed in some cases. Yet, stress responses are typically mounted within minutes. The mechanical response to the laser stimulus, the chromatin movement, can be seen independent as it is much faster. Therefore, mechanical properties of the chromatin movement are independent of a later stress response. To dissect the occasional stress response, further studies would be needed, which could benefit from decoupling of laser heating and the induction of temperature gradients (Minopoli et al., 2023).”

The three additional references, Hofmann et al., 2021, Sidrauski et al., 2015, and Minopoli et al., 2023, have been added to the reference list of the manuscript.

Further, we had provided several arguments in the original rebuttal letter why we think that the short-lived absolute temperature increase above 36°C does not exert a considerable stress. The newly revised manuscript is now supplemented with such arguments in the beginning of the Discussion section.

“Concerning the slight and short-lived temperature increase above 36°C during our perturbation, we would like to mention that the range of temperature fluctuations that naturally occur within tissues and cells of organisms or that cells experience during common experimental practices are much wider than commonly assumed. For example, the arguable most temperature sensitive cells are fertilized human egg cells, and considerable efforts are made to ensure highest temperature stability. Yet, the actual temperature in hoods and on microscope surfaces is in average more than 1.3°C different from the displayed temperature (Palmer et al., 2019). Also, core temperature in mice is not really constant but may vary by up to 5°C depending on ambient temperature (Kaplan and Leveille, 1974). Bovine embryonic development in vitro was unaffected by temperatures of up to 40°C, while deleterious effects were observable at a temperature of 41°C (Rivera and Hansen, 2001). This all is against the common view that only 37°C are seen as physiological temperature for mammalian cells.”

The three additional references, Palmer et al., 2019, Kaplan and Leveille, 1974, and Rivera and Hansen, 2001 have been added to the reference list of the manuscript.

Furthermore, we had already eliminated the word “non-invasive” from the title of the manuscript.

We now removed also all four remaining occurrences of the term “non-invasive” as a characteristic of our method. The occurrences were in:

– Main findings, first point

– Introduction, last paragraph

– Discussion, first paragraph

– Discussion, first paragraph of the second part

3. A key point in the rebuttal was the statement that "…we would like to emphasise that we do not think the laser itself nor the short temperature increase up to 39{degree sign}C exert a considerable stress or produce a considerable strain…It is the sharp temperature gradient, created by the scan pattern that we apply, that induces the chromatin displacement (similar to the gradient in an electric potential causing the migration of ions in electrophoresis, rather than the absolute magnitude of the potential)." In addition to including the author's perspective on this point in the text of the paper, not just the rebuttal, we also ask that the support for this statement is expanded in the manuscript. For example, a quantitative physical argument or experimental evidence for this point needs to be provided, particularly as it lies at the heart of the new method.

The reviewers highlighted our statement that the chromatin movement is not caused by slightly elevated absolute temperature upon the laser stimulus, but the temperature gradient.

We had explained our reasoning in the previous rebuttal letter (albeit a bit scattered), and had added additional experimental evidence, but have not provided extensive explanation in the revised manuscript. Instead of just referencing the supplementary figure showing our experiments at 30°C instead of 36°C, we now elaborate on it more.

We have included a paragraph into the first results chapter which explains our experiments at 30°C and provides an argument for an absolute temperature slightly above 36°C not being required and therefore not causative of the chromatin movement.

“To exclude the possibility that an absolute temperature increase above 36°C is the driver of the observed chromatin movement, rather than the induced temperature gradient, we repeated our experiments at an ambient temperature of 30°C. As with an ambient temperature of 36°C, we observed chromatin motion upon the heating stimulus down the temperature gradient (Figure 1—figure supplement 1c-e).”

Further, we have included a paragraph into the beginning of the discussion providing additional arguments.

“We have shown that chromatin motion down the laser induced temperature gradient occurs also at ambient temperatures much lower than 36°C, suggesting that the driver of the chromatin motion is the temperature gradient rather than the absolute temperature. From a rigorous physics point of view, one should note that a temperature gradient has a direction (is vectorial) which was consistently observed to be parallel to the observed motion of chromatin. The mere rise of temperature constitutes a rise of a scalar quantity, which does not provide a direction that could explain the directed motion of chromatin. Hence, it should be noted that only the gradient and not the absolute rise in temperature falls into a class of symmetries that is suitable to account for the observed effect of chromatin motion.”